# SmartPretrain: Model-Agnostic and Dataset-Agnostic Representation Learning for Motion Prediction

**Yang Zhou[1][*]  Hao Shao[1,2][*†]  Letian Wang[3][*]**
**Steven L. Waslander[3]  Hongsheng Li[2,4,5]  Yu Liu[1,5]**

[1]SenseTime Research  [2]CUHK MMLab  [3]University of Toronto
[4]CPII under InnoHK  [5]Shanghai Artificial Intelligence Laboratory

## Abstract

Predicting the future motion of surrounding agents is essential for autonomous vehicles (AVs) to operate safely in dynamic, human-robot-mixed environments. However, the scarcity of large-scale driving datasets has hindered the development of robust and generalizable motion prediction models, limiting their ability to capture complex interactions and road geometries. Inspired by recent advances in natural language processing (NLP) and computer vision (CV), self-supervised learning (SSL) has gained significant attention in the motion prediction community for learning rich and transferable scene representations. Nonetheless, existing pre-training methods for motion prediction have largely focused on specific model architectures and single dataset, limiting their scalability and generalizability. To address these challenges, we propose SmartPretrain, a general and scalable SSL framework for motion prediction that is both model-agnostic and dataset-agnostic. Our approach integrates contrastive and reconstructive SSL, leveraging the strengths of both generative and discriminative paradigms to effectively represent spatiotemporal evolution and interactions without imposing architectural constraints. Additionally, SmartPretrain employs a dataset-agnostic scenario sampling strategy that integrates multiple datasets, enhancing data volume, diversity, and robustness. Extensive experiments on multiple datasets demonstrate that SmartPretrain consistently improves the performance of state-of-the-art prediction models across datasets, data splits and main metrics. For instance, SmartPretrain significantly reduces the MissRate of Forecast-MAE by 10.6%. These results highlight SmartPretrain's effectiveness as a unified, scalable solution for motion prediction, breaking free from the limitations of the small-data regime. The code is available at https://github.com/youngzhou1999/SmartPretrain.

## 1 Introduction

Motion prediction, predicting the future states of space-sharing agents nearby (*e.g.*, vehicle, cyclist, pedestrian) is crucial for autonomous driving systems (Hu et al., 2023; Shao et al., 2023b;a; 2024) to safely and efficiently operate in the dynamic and human-robot-mixed environment. Context information, including surrounding agents' states and high-definition maps (HD maps), provides critical geometric and semantic information for motion behavior, as agents' behaviors are highly dependent on interactions with surrounding agents and the map topology. For example, agents' interactive cues, such as yielding, would influence other agents' decision-making, and vehicles usually move in drivable areas and follow the direction of lanes. Thus, designing (Chai et al., 2019a; Gao et al., 2020; Liang et al., 2020; Wang et al., 2022) and learning (Salzmann et al., 2020; Ngiam et al., 2021; Varadarajan et al., 2022; Zhao et al., 2021) scene representation that captures rich motion and context information has long been a core challenge for motion prediction.

---

[*]Equal contribution.
[†]Project lead.

However, while natural language processing (NLP) and computer vision (CV) communities have repeatedly demonstrated the power of large-scale datasets, existing motion prediction works have been operating in a "small-data regime", due to the expensive and laborious nature of collecting and annotating driving trajectory data. For example, popular motion prediction datasets such as the Argoverse (Chang et al., 2019), Argoverse 2 (Wilson et al., 2023) and Waymo Open Motion Dataset (WOMD) (Sun et al., 2020) datasets only include 320K, 250K, 480K data sequences, respectively, significantly less than the common-practice data scale in the NLP and CV community (e.g., ChatGPT-3 trained on the 400B-token CommonCrawl dataset (Raffel et al., 2020), Vision Transformer trained on the 303M-image JFT dataset (Sun et al., 2017)). The scarcity of trajectory data prohibits the models from learning rich and transferable scene representation, and thus restrains their performance and generalizability. To this end, one line of work has sought to mitigate the scarcity of real driving motion data, by leveraging synthetic trajectory data generated via prior knowledge and manually designed rules (Li et al., 2024; Azevedo et al., 2022). However, a major drawback of using synthetic data is the "reality gap" — the distribution of artificially generated data often differs from real-world distributions, leading to a gap between synthetic training and real-world performance. Additionally, generating high-fidelity simulations requires considerable computational resources and careful design to capture complex agent interactions accurately, limiting scalability.

In addition to data scaling, the NLP and CV communities have developed remarkable advances in self-supervised learning (SSL). As evidenced by the success of models such as BERT (Devlin et al., 2019) and Masked Autoencoders (He et al., 2022), SSL is demonstrated to acquire expressive representations/features by pre-training on unlabelled data, and thus enhances downstream performance after fine-tuning. To this end, SSL has recently received increased attention in the motion prediction community. However, designing truly general and scalable SSL pre-training strategies for motion prediction has been non-trivial. We elaborate on the challenges by categorizing existing works into two primary approaches: generative SSL and discriminative SSL.

- In the domain of generative SSL, pretext tasks are designed to learn context-rich scene representations from real data. Various traffic-related pretext tasks have been proposed, such as classification for maneuver or forecasting success (Bhattacharyya et al., 2022), map graph relationship prediction (Azevedo et al., 2022), and traffic event detection (Pourkeshavarz et al., 2023). Moreover, extensive efforts have been made to replicate the success of Masked Autoencoder techniques seen in NLP and CV, exploring diverse traffic-specific masking strategies (Yang et al., 2023; Chen et al., 2023; Lan et al., 2024; Cheng et al., 2023). However, these pre-training strategies usually impose restrictions on the model architecture - they start by proposing pre-training strategies, whose realization then necessitates the model to generate certain types of features. For example, the detection for maneuvering, graph relationships, and traffic events require explicit formulation of these concepts/features within the model. MAE approaches demand that each trajectory or map segment must have an explicit feature representation to enable reconstructive pre-training. In approaches where only trajectories are masked and reconstructed (Yang et al., 2023), the method remains simple and flexible but essentially acts as a data augmentation technique. While many works focus on aggregating agent embeddings and provide explicit access to them, explicit map embeddings are not always available (Zhou et al., 2023; Tang et al., 2024). Thus the map reconstruction pre-training strategy in MAE could be less general. Due to such inflexibility, these SSL strategies can usually only apply to specific models or, in most cases, their single specific model.

- In discriminative SSL, contrastive learning has emerged as a promising technique for motion prediction tasks, where methods aim to learn discriminative features by contrasting the trajectory and map embeddings among positive and negative examples (Xu et al., 2022). However, it is limited to models based on rasterized map representations, which is shown to have a significant performance gap compared to more recent transformer-based or graph-based models that incorporate vectorized map (Liang et al., 2020; Gao et al., 2020). Applying CL to vectorized representations remains an under-explored area, due to the irregular structure of vector data, which complicates the definition of meaningful positive and negative samples and requires sophisticated sampling strategies.

Summing up, due to the inherently complex and multimodal input representation for motion prediction, existing SSL pre-training strategies fall short in their applicability to general model architectures and input representations. Moreover, the discrepancies among different popular motion prediction datasets have further prevented these methods from leveraging multiple datasets to break free from the small-data regime. As shown in Fig. 1, such limitations detract from the true power

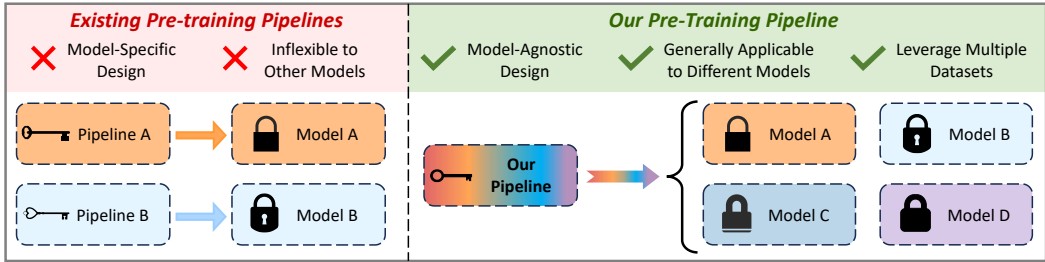

Figure 1: Illustration comparing existing trajectory prediction pre-training pipelines with ours. Our pipeline can unlock performance gains from all models as it is model-agnostic, while most existing pipelines are model-specific and inflexible.

of SSL, which lies in its ability to provide a unified and scalable strategy for representation learning across diverse model architectures, input representation, and data sources.

In this paper, we propose *SmartPretrain*, a general and scalable SSL representation learning approach for motion prediction. To fully unleash the power of SSL, SmartPretrain is specifically designed to be model-agnostic and dataset-agnostic, providing a unified solution that is broadly applicable across different model architectures and diverse data sources, thus escaping from the suboptimal representation learning of the small-data regime. Our key designs are twofold:

- *Model-Agnostic Contrastive and Reconstructive SSL*: SmartPretrain proposes a novel model-agnostic SSL framework, which combines the best of the two worlds of generative and discrimitive SSL. In a nutshell, we simultaneously reconstruct trajectories and contrast trajectory embeddings from different agents and time windows, to learn spatiotemporal evolution and interaction happening in the scene. These trajectory-focused SSL pretext tasks avoid restrictions on the model architecture and map representation, making SmartPretrain adaptable to a wide range of model designs, whether raster-based, transformer-based or graph-based.

- *Dataset-Agnostic Scenario Sampling*: We propose a dataset-agnostic scenario sampling strategy that scales effectively by integrating multiple datasets, despite inherent discrepancies. To achieve this, we standardize data representations, ensure high-quality inputs, and maximize data volume and diversity. This allows SmartPretrain to leverage a broader range of driving scenarios, enhancing generalization and robustness.

Extensive experiments on applying SmartPretrain to multiple state-of-the-art prediction models and multiple datasets show that, SmarPretrain consistently improves all considered models, on downstream datasets, data splits and main metrics. For instance, SmartPretrain significantly reduces the minFDE, minADE, MR of QCNet by 4.9%, 3.3%, 7.6%, and Forecast-MAE by 4.5%, 3.1%, 10.6%. SmartPretrain exhibits superior performance when compared to existing pre-training methods. Comprehensive ablation studies are conducted to analyze each component of our pipeline, such as scaling with multiple datasets, and the effect of the two proposed pretext tasks. To the best of our knowledge, SmartPretrain is the very first work that conducts SSL pre-training leveraging multiple datasets and can be applied to a range of models, for motion prediction in the driving domain.

## 2 RELATED WORK

### 2.1 TRAJECTORY PREDICTION

Traditional methods for motion prediction primarily use Kalman filtering (Kalman & Others, 1960) with physics and maneuver priors from HD-maps to predict future motion states (Houenou et al., 2013; Xie et al., 2017; Shao et al., 2023a), or sampling-or-optimization-based planning algorithms with manually specified or learned reward functions to generate future trajectories (Wang et al., 2021; 2023a; Schwarting et al., 2019; Li et al., 2022). With the rapid development of deep learning, recent works utilize data-driven approaches for motion prediction. Generally, these methods fit into three different architectures involving rasterized images and CNNs (Cui et al., 2019; Chai et al., 2019b), vectorized representations and GNNs (Zhou et al., 2022; 2023; Park et al., 2023; Tang et al., 2024) and transformers (Wang et al., 2023b; Nayakanti et al., 2023; Shi et al., 2024). To represent scene information, raster-based methods utilize CNNs to rasterize scene context into a

bird-eye-view image, while the other two architectures represent each entity of the scene as a vector following Gao et al. (2020). For the multimodal trajectory outputs, in addition to the standard one-stage prediction pipelines which output trajectories directly, there are now two-stage prediction-refinement pipelines (Ye et al., 2023; Choi et al., 2023; Zhou et al., 2023; Shi et al., 2024; Zhou et al., 2024) where coarse trajectories are first proposed and then used for trajectory refinement.

## 2.2 SELF-SUPERVISED LEARNING IN TRAJECTORY PREDICTION

Self-supervised learning (SSL) methods have been applied widely in both visual understanding (Chen et al., 2020; Grill et al., 2020; Caron et al., 2021; He et al., 2022; Benaim et al., 2020; Bertasius et al., 2021; Feichtenhofer et al., 2022; Lin et al., 2023; 2024; Tang et al., 2025), natural language processing (Devlin et al., 2019; Brown, 2020; Touvron et al., 2023) and multimodal representation learning (Qu et al., 2025). SSL aims to learn informative and general representations via carefully designed pretext tasks, which can be fine-tuned for downstream tasks with supervision. Recently, the trajectory prediction community has made significant strides in incorporating SSL techniques. These methods utilize pretext tasks to pre-train models, allowing them to learn valuable representations that can be fine-tuned for enhanced trajectory prediction performance. Existing pre-training pipelines for trajectory prediction can be divided into three different categories, involving augmented or synthetic data (Yang et al., 2023; Li et al., 2024), contrastive learning (Xu et al., 2022; Bhattacharyya et al., 2022; Azevedo et al., 2022; Pourkeshavarz et al., 2023) and generative masked representation learning (Chen et al., 2023; Cheng et al., 2023; Lan et al., 2024).

Methods with synthetic data generate trajectory and map data with prior knowledge and manually designed rules. For example, Li et al. (2024) generates driving scenarios with a map augmentation module and a model-based planning model. Contrastive learning methods aim to align and differentiate embeddings by comparing them with positive and negative examples, capturing high-level semantic relationships. For example, PreTraM (Xu et al., 2022), a raster-based approach, uses contrastive learning to model the relationships between trajectories and maps, as well as between the maps themselves. On the other hand, generative masked representation methods leverage the transformer (Vaswani et al., 2023) to learn token relationships by reconstructing randomly masked HD map context and trajectories, inspired by the broad success of Masked Autoencoders (MAE) (He et al., 2022). For example, Forecast-MAE (Cheng et al., 2023) treats agents' histories and lane segments as individual tokens, applies the random mask at the token level, and feeds the masked tokens into the transformer backbone for reconstruction.

Existing SSL pipelines have two major limitations: 1) they pre-train on a single trajectory prediction dataset due to the varied formats in different datasets and 2) they rely on specific model architectures and map embeddings, limiting their adaptability to general models. Additionally, these pipelines struggle to extend to advanced GNN-based approaches, which integrate all inputs into a unified graph, preventing the use of explicit map embeddings for pretext tasks. To overcome these challenges, we propose SmartPretrain, a model-agnostic and dataset-agnostic solution. It can flexibly apply to various models and datasets, regardless of model architecture or dataset format.

## 3 METHODOLOGY

### 3.1 PROBLEM FORMULATION - MOTION PREDICTION MEETS SELF-SUPERVISED LEARNING

Typical motion prediction task can be defined as follows: taking the observed states of the target agent $\mathbf{s}_h = [s_{-T_h+1}, s_{-T_h+2}, \ldots, s_0] \in \mathbb{R}^{T_h \times 2}$ over the last $T_h$ steps, we aim to predict its future states $\mathbf{s}_f = [s_1, s_2, \ldots, s_{T_f}] \in \mathbb{R}^{T_f \times 2}$ for $T_f$ future steps as well as the associated probabilities $\mathbf{p}$. Naturally, the target agent will interact with its context $\mathbf{c}$, including observed states of surrounding agents, and the HD map. The typical motion prediction task is formulated as $(\mathbf{s}_f, \mathbf{p}) = f(\mathbf{s}_h, \mathbf{c})$, where $f$ denotes the prediction model. Generally, motion prediction models are structured in an encoder-decoder architecture with two stages:

- *Encoding* $\mathbf{z} = f_{enc}(\mathbf{s}_h, \mathbf{c})$: an encoder $f_{enc}$ embeds and fuses $\mathbf{s}_h$ and $\mathbf{c}$ to capture the evolution and interaction in the traffic scene, and generate trajectory embedding $\mathbf{z}$;
- *Decoding* $(\mathbf{s}_f, \mathbf{p}) = f_{dec}(\mathbf{z})$: a decoder $f_{dec}$ decipher $\mathbf{z}$ to generates multiple possible future trajectories $\mathbf{s}_f$ and corresponding probabilities $\mathbf{p}$.

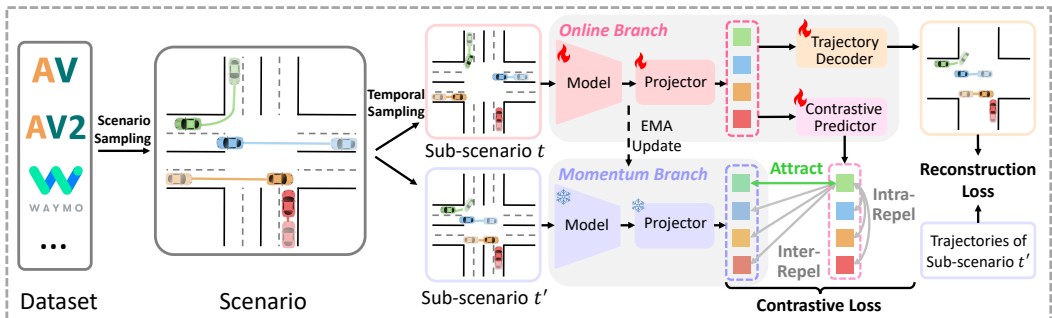

Figure 2: Overview of our model-agnostic and dataset-agnostic pre-training pipeline. We begin by randomly sampling a training scenario from mixed datasets. From this scenario, two sub-scenarios with different temporal timelines are randomly sampled and fed into two model branches to generate trajectory embedding in the scene. Two model-agnostic pretext tasks, trajectory contrastive learning and reconstructive learning, are introduced to learn transferable and robust representations.

SSL takes a pivot from the typical motion prediction learning by introducing a self-supervised pre-training phase. The objective is to pre-train the encoder $f_{enc}$ with pretext tasks, to learn more transferable and generalized embeddings $\mathbf{z}$. Afterward, the encoder $f_{enc}$ and decoder $f_{dec}$ are fine-tuned together on the actual motion prediction task. In the literature, the challenges and focus have been on designing pretext tasks that scale well and effectively enhance downstream performance.

## 3.2 SMARTPRETRAIN SSL FRAMEWORK

We propose SmartPretrain, a model-agnostic and dataset-agnostic pre-training framework for motion prediction, that can be flexibly applied to a range of models regardless of their architectures, and leverage various datasets despite their differences in data formats. As illustrated in Fig. 2, Smart-Pretrain is composed of two parts: 1) a dataset-agnostic scenario sampling strategy for constructing representative and diverse pre-training data, and 2) a model-agnostic SSL strategy, consisting of two trajectory-focused pretext tasks, trajectory contrastive learning (TCL) and trajectory reconstruction learning (TRL), which together shape the pre-training to improve performance on downstream tasks.

### 3.2.1 DATASET-AGNOSTIC SAMPLING

In the dataset-agnostic sampling process, we aim to create positive and negative pairs for contrastive learning and construct trajectories for reconstructive learning, while ensuring data consistency across various datasets.

**Data Sampling for Contrastive/Reconstructive SSL.** Starting with formulating positive and negative samples for contrastive learning, one intuitive design would be contrasting different agents' trajectories within the same scenario. However, this approach may lead to suboptimal performance due to 1) an overemphasis on learning spatial context among agents without sufficient temporal modeling and 2) limited variability in positive samples, as they contain the same features. To this end, we propose a temporal sampling strategy to create sample pairs that capture both the spatial context and the temporal evolution of the agents. Specifically, we mix multiple datasets to form a comprehensive data bank, and randomly sample a scenario with a time horizon $T = T_h + T_f$. Next, we temporally sample two sub-scenarios, which have the same temporal horizon $T_h$, but start at different time $t$ and $t'$ respectively. To prevent information leakage during sub-scenario sampling, we ensure that the two sub-scenarios of a single scenario do not overlap, which could compromise the training of the pretext tasks. The two sub-scenarios are then later used to construct a positive trajectory pair from the same agent across different time, and negative pairs from different agents or different time. Note that, beyond contrastive learning, the sub-scenario starting at $t$ will also be used for reconstructive learning, as its temporal horizon is designed to align well with the input horizon $T_h$ of the target downstream dataset.

**Maintaining Dataset-Agnosticism.** To leverage multiple datasets with varied configurations and achieve data scaling, we introduce three key designs:

- *Standardizing Representations:* Due to varying formats across datasets, HD map's resolution and agents' trajectory horizon often differ significantly, causing inconsistency. To address this, we standardize the map contexts and trajectories representation into a unified format. Specifically, we fix the resolution of HD maps using linear interpolation or downsampling to ensure consistency between consecutive points. Additionally, to handle differences in scenario horizons, we apply zero-padding to scenarios with shorter horizons to align their length with the desired time horizon $T$. The padded trajectory steps are masked as invalid during pre-training to ensure they do not affect the learning process.

- *Ensuring Data Quality:* Different datasets also vary in data quality and scenario complexity, often containing incomplete trajectories due to perception limitations or agents' entering/leaving the scene. To improve consistency and ensure higher data quality, we include only complete trajectories during pre-training, filtering out incomplete ones from the training pipeline.

- *Maximizing Data Volume and Diversity:* The number of scenarios varies significantly among datasets. We found that directly mixing all available training data, rather than balancing the number of trajectories from each dataset, provides the best downstream results.

### 3.2.2 Model-Agnostic Contrastive and Reconstructive SSL

We propose a model-agnostic SSL strategy that consists of two pretext tasks: 1) a trajectory contrastive learning task (TCL), that enriches learned trajectory embeddings by contrasting them across agents and time windows; 2) a trajectory reconstruction learning task (TRL), that aligns more closely with the primary goal of motion prediction, better shaping the direction of pre-training. Note that both pretext tasks are designed to ensure *model-agnosticism* via their trajectory-focus: they only contrast and reconstruct agents' trajectory embeddings, rather than any other embeddings such as map or customized embeddings. Thus they remove restrictions on the model architectures and map representations, and can be applied to a much wider variety of motion prediction models.

**Embedding Generation.** With 2 sampled sub-scenarios, we then generate embeddings for all the trajectories they contain. Specifically, we follow a self-training strategy from existing SSL literature (He et al., 2020; Caron et al., 2021) to bootstrap performance and avoid model collapse. Specifically, the two sub-scenarios are fed into two identical architecture branches, an online branch and a momentum branch. Within each branch, the input sub-scenario is first passed through the motion prediction model to generate all trajectories' embeddings, which are then fed to a projector for further encoding modification. During pre-training, the online branch is continuously updated, while the momentum branch is occasionally updated using an exponential moving average mechanism. The embeddings from the online branch are also additionally passed through a contrastive predictor to generate the final embeddings for contrastive learning.

**Trajectory Contrastive Learning (TCL).** TCL is designed to learn rich trajectory representations by contrasting trajectory embeddings across spatiotemporal dimensions. Specifically, considering that the sampled scenarios in a mini-batch consist of $N$ agents, we now have two sets of embeddings $\{\mathbf{z}_{i,t}\}_{i=1}^{N}$ and $\{\mathbf{z}'_{j,t'}\}_{j=1}^{N}$, generated from the online branch and momentum branch respectively. We define the contrastive loss to pull closer positive samples, and repel away negative samples:

$$\mathcal{L}_c = -\frac{1}{N}\sum_{i=1}^{N}\log\frac{\exp\left(r(\mathbf{z}_{i,t}, \mathbf{z}'_{i,t'})/\tau\right)}{\sum_{j=1,j\neq i}^{N}\exp\left(r(\mathbf{z}_{i,t}, \mathbf{z}_{j,t})/\tau\right) + \sum_{j=1}^{N}\exp\left(r(\mathbf{z}_{i,t}, \mathbf{z}'_{j,t'})/\tau\right)}, \quad (1)$$

where $r$ denotes the cosine similarity measure, and $\tau$ denotes the temperature hyper-parameter. Specifically, the numerator term represents the similarity of positive sample pairs $(\mathbf{z}_{i,t}, \mathbf{z}'_{i,t'})$, namely trajectory embeddings from the same agent in different timelines. The denominator term consists of two types of negative pairs: 1) intra-repelling pairs $(\mathbf{z}_{i,t}, \mathbf{z}_{j,t})$: the trajectory embeddings from other agents of the same sub-scenario; 2) inter-repelling pairs $(\mathbf{z}_{i,t}, \mathbf{z}'_{j,t'})$: trajectory embeddings from different sub-scenarios. Through this objective, the similarities of the same agent's embeddings are maximized, and those of the other pairs are minimized, thereby refining the model's ability to capture meaningful contextual relationships and temporal dynamics.

**Trajectory Reconstruction Learning (TRL).** While contrastive learning is a discriminative task that helps features distinguish motion and contextual differences among trajectories, the ultimate goal of motion prediction is a regression task. Therefore, the features learned only through contrastive learning may not necessarily align with or benefit the needs of motion prediction. To this

end, we propose trajectory reconstruction learning, which is designed to bring the learned representation as close as possible to the needs of motion prediction, by sharing a similar training objective.

Specifically, recall that in Sec. 3.2.1, we sampled a scenario with horizon length $T$, and generated the trajectory embeddings $\{\mathbf{z}_{i,t}\}_{i=1}^N$ for sub-scenario with horizon length $T_h$ and start time $t$. We then pass $\{\mathbf{z}_{i,t}\}_{i=1}^N$ to a trajectory decoder, to reconstruct the trajectory segments from the sub-scenario with starting time $t'$. The reconstruction loss $\mathcal{L}_r$ is then simply designed as the average $L_1$ distance between the reconstructed trajectories and the ground truth. Interestingly, the proposed reconstruction loss can be viewed as a generalized form of regression loss for the downstream trajectory prediction task. When $t$ equals 0, the input segment is exactly the observed states $\mathbf{s}_h$ of the downstream trajectory prediction task. The task regresses the trajectory with $T_h$ steps started from $t'$ of the future states $\mathbf{s}_f$, based on the observed states $\mathbf{s}_h$.

### 3.2.3 TRAINING DETAILS

**Pre-training Stage.** The overall pre-training scheme integrates trajectory contrastive learning and reconstruction. The combined loss function is formulated as follows:

$$\mathcal{L} = \mathcal{L}_c + \lambda \mathcal{L}_r, \tag{2}$$

where $\lambda$ is a hyper-parameter balancing the contribution of both tasks and we set $\lambda$ as 1.0.

**Finetuning Stage.** After pre-training on a specific model, we initialize the model's encoder $f_{enc}$ with pre-trained weights, and fine-tune the whole model on the downstream trajectory prediction task, using the model's original prediction objective and training schedules.

## 4 EXPERIMENTS

### 4.1 EXPERIMENTAL SETUP

**Datasets.** We train and evaluate our method on three large-scale motion forecasting datasets: Argoverse (Chang et al., 2019), Argoverse 2 (Wilson et al., 2023) and Waymo Open Motion Dataset (WOMD) (Sun et al., 2020). Argoverse contains 333k scenarios collected from interactive and dense traffic. Each scenario provides the HD map and 2 seconds of historic trajectory data, to predict the trajectory for the next 3 seconds, sampled at 10Hz. The training, validation, and test set of Argoverse is set to 205k, 39k and 78k scenarios, respectively. Argoverse 2 extends the historic and prediction horizon to 5 seconds and 6 seconds respectively, sampled at 10Hz. The data is split into 200k, 25k, and 25k for training, validation, and test, respectively. For WOMD, the dataset provides 1 second of historic trajectory data and aims to predict the trajectory for 8 seconds into the future, sampled at 10Hz as well. It contains 487k training scenes, 44k validation scenes and 44k testing scenes. Notably, our pre-training pipeline only utilizes the training splits of these datasets.

**Baselines.** As mentioned previously, our pre-training pipeline can be seamlessly integrated into most existing trajectory prediction methods. In our experiments, we consider four popular and advanced methods as the prediction backbone to evaluate how our SmartPretrain further improves performance: HiVT (Zhou et al., 2022), HPNet (Tang et al., 2024), Forecast-MAE (Cheng et al., 2023) and QCNet (Zhou et al., 2023). We use their official open-sourced code for implementation.

**Metrics.** Following the official dataset settings (Chang et al., 2019), we evaluate our model using the standard metrics for motion prediction, including minimum Average Displacement Error (minADE), minimum Final Displacement Error (minFDE), and Miss Rate (MR). While the prediction model forecasts up to 6 trajectories for each agent, these metrics evaluate the trajectory with the minimum endpoint error, as a signal of best possible performance from the multi-modal predictions.

**Implementation Details.** We implement the projector and contrastive predictor as a 2-layer MLP with batch normalization, and the trajectory decoder also as a 2-layer MLP but with layer normalization to better fit its sequence nature. We use AdamW to optimize the online branch. In the momentum branch, the weights of the motion prediction encoder and the projector are initialized to be identical to that of the online branch, and updated via an exponential moving average (EMA) strategy. We use a momentum value of 0.996 and increase this value to 1.0 with a cosine schedule. We conduct single dataset pre-training with 8 Nvidia A100 40GB GPUs and data-scaled pertaining with 32 GPUs, both for 128 epochs. For fine-tuning, we use 8 GPUs with the models' original training schedules.

| Downstream Dataset | Method | Pre-train Dataset | Validation Set | | | Test Set | | |
|---|---|---|---|---|---|---|---|---|
| | | | minFDE ↓ | minADE ↓ | MR ↓ | minFDE ↓ | minADE ↓ | MR ↓ |
| Argoverse | HiVT | - | 0.969 | 0.661 | 0.092 | 1.169 | 0.774 | 0.127 |
| | HiVT w/ Ours | Argo | 0.940 (-3.0%) | 0.647 (-2.1%) | 0.088 (-4.3%) | 1.146 (-2.0%) | 0.763 (-1.4%) | 0.122 (-3.9%) |
| | HiVT w/ Ours | All | 0.929 (-4.1%) | 0.644 (-2.6%) | 0.086 (-6.5%) | 1.135 (-2.9%) | 0.760 (-1.8%) | 0.119 (-6.3%) |
| | HPNet | - | 0.871 | 0.638 | 0.069 | 1.099 | 0.761 | 0.107 |
| | HPNet w/ Ours | Argo | 0.854 (-2.0%) | 0.631 (-1.1%) | 0.065 (-5.8%) | 1.090 (-0.8%) | 0.758 (-0.4%) | 0.103 (-3.7%) |
| Argoverse 2 | QCNet | - | 1.253 | 0.720 | 0.157 | 1.241 | 0.636 | 0.154 |
| | QCNet w/ Ours | Argo 2 | 1.212 (-3.3%) | 0.702 (-2.5%) | 0.148 (-5.8%) | 1.222 (-1.5%) | 0.625 (-1.7%) | 0.146 (-5.2%) |
| | QCNet w/ Ours | All | 1.191 (-4.9%) | 0.696 (-3.3%) | 0.145 (-7.6%) | 1.203 (-3.1%) | 0.618 (-2.8%) | 0.142 (-7.8%) |
| | Forecast-MAE | - | 1.436 | 0.811 | 0.189 | 1.427 | 0.727 | 0.187 |
| | Forecast-MAE w/ Ours | Argo 2 | 1.372 (-4.5%) | 0.786 (-3.1%) | 0.169 (-10.6%) | 1.390 (-2.6%) | 0.713 (-1.9%) | 0.171 (-8.5%) |

Table 1: We evaluate the performance of SmartPretrain with multiple state-of-the-art prediction models, reporting results on the validation and test sets of the Argoverse and Argoverse 2 datasets. SmartPretrain consistently enhances all models across datasets, data splits and main metrics. Additionally, pre-training with more datasets results in further performance improvements.

| Dataset | Backbone | Pre-training Method | minFDE ↓ | minADE ↓ | MR ↓ |
|---|---|---|---|---|---|
| Argoverse 2 | Forecast-MAE | Without Pre-training | 1.436 | 0.811 | 0.189 |
| | | Forecast-MAE | 1.409 (-1.9%) | 0.801 (-1.2%) | 0.178 (-6.2%) |
| | | Ours | **1.372** (-4.5%) | **0.786** (-3.1%) | **0.169** (-10.6%) |

Table 2: Performance comparison with existing pre-training method Forecast-MAE.

## 4.2 QUANTITATIVE RESULTS

**Performance of Applying SmartPretrain to Multiple Models.** As shown in Table 1, we first report the performance when we apply SmartPretrain to multiple state-of-the-art prediction models, on the validation and test set of Argoverse and Argoverse 2. We consider two pre-training settings: pre-training only on the single downstream dataset, and pre-training on all three datasets. Specifically, we pre-train HiVT and QCNet with both two settings, and only pre-train HPNet and Forecast-MAE with the single downstream dataset due to compute constraints. SmartPretrain can consistently improve all considered models, on downstream datasets, data splits and main metrics. For instance, SmartPretrain can significantly reduce the minFDE, minADE, MR of QCNet on validation set by 4.9%, 3.3%, 7.6% respectively. Besides, Pre-training with all datasets also shows consistently higher improvement compared to pre-training with only one dataset.

**Performance Comparison with other Pre-training Method.** We also compare SmarPretrain with other pre-training methods. For a fair and meaningful comparison, we look for methods that have open-source code. However, to the best of our knowledge, only Forecast-MAE was open-sourced at the time of this paper's submission. Specifically, Forecast-MAE proposes a motion prediction backbone and a pre-training strategy. We then apply SmartPretrain to Forecast-MAE's backbone, and compare it with Forecast-MAE's pre-training strategy, on the Argoverse 2 dataset. As in Table 2, our pre-training method shows a larger improvement than Forecast-MAE's pre-training method by a substantial margin (*e.g.* improvement of 4.5% vs 1.9% on minFDE).

These results demonstrate that our pre-training pipeline: 1) can be flexibly applied to a wide range of motion prediction models; 2) consistently improves performance through pre-training and data scaling; and 3) delivers stronger performance enhancements compared to existing pre-training methods.

## 4.3 ABLATION STUDIES

We conduct comprehensive ablation studies to analyze the impact of different components, including the data scale of pre-training, the two proposed pretext tasks, and associated hyperparameters and configurations. For efficient evaluation, we use HiVT as the prediction model and Argoverse for both pre-training and fine-tuning, reporting performance on the Argoverse validation set.

**Ablations on Pre-training Datasets.** SmartPretrain is designed to be dataset-agnostic and thus can leverage multiple datasets for pre-training. Here we introduce four pre-training settings: 1) *No Pre-Training*: the model is trained from scratch on the downstream task without pre-training; 2) *Baseline Pre-Training*: the model is pre-trained on a single dataset, which is the same as the downstream dataset; 3) *Transfer Pre-Training*: the model is pre-trained on a single dataset different from the downstream dataset; 4) *Data-Scaled Pre-Training*: the model is pre-trained on multiple

| HiVT on | Pre-training Dataset | | | Validation Set | | |
|---|---|---|---|---|---|---|
| Argoverse | Argoverse | Argoverse 2 | WOMD | minFDE ↓ | minADE ↓ | MR ↓ |
| No Pre-training | ✗ | ✗ | ✗ | 0.969 | 0.661 | 0.092 |
| Baseline Pre-training | ✔ | ✗ | ✗ | 0.940 (-3.0%) | 0.647 (-2.1%) | 0.088 (-4.3%) |
| Transfer Pre-training | ✗ | ✔ | ✗ | 0.951 (-1.9%) | 0.653 (-1.2%) | 0.090 (-2.2%) |
| | ✗ | ✗ | ✔ | 0.946 (-2.4%) | 0.652 (-1.4%) | 0.089 (-3.3%) |
| Data-Scaled Pre-training | ✔ | ✔ | ✗ | 0.935 (-3.5%) | 0.645 (-2.4%) | 0.087 (-5.4%) |
| | ✔ | ✗ | ✔ | 0.935 (-3.5%) | 0.645 (-2.4%) | **0.086** (-5.4%) |
| | ✔ | ✔ | ✔ | **0.929** (-4.1%) | **0.644** (-2.6%) | **0.086** (-6.5%) |
| QCNet on | Pre-training Dataset | | | Validation Set | | |
| Argoverse 2 | Argoverse | Argoverse 2 | WOMD | minFDE ↓ | minADE ↓ | MR ↓ |
| No Pre-training | ✗ | ✗ | ✗ | 1.253 | 0.720 | 0.157 |
| Baseline Pre-training | ✗ | ✔ | ✗ | 1.212 (-3.3%) | 0.702 (-2.5%) | 0.148 (-5.7%) |
| Transfer Pre-training | ✔ | ✗ | ✗ | 1.226 (-2.2%) | 0.712 (-1.1%) | 0.151 (-3.8%) |
| | ✗ | ✗ | ✔ | 1.225 (-2.2%) | 0.708 (-1.7%) | 0.149 (-5.1%) |
| Data-Scaled Pre-training | ✔ | ✔ | ✗ | 1.201 (-4.2%) | 0.702 (-2.5%) | **0.143** (-8.9%) |
| | ✗ | ✔ | ✔ | 1.199 (-4.3%) | 0.698 (-3.1%) | 0.145 (-7.6%) |
| | ✔ | ✔ | ✔ | **1.191** (-4.9%) | **0.696** (-3.3%) | 0.145 (-7.6%) |

Table 3: Ablations on Pre-training Datasets: All pre-training strategies effectively improve prediction accuracy compared to training from scratch. However, transfer pre-training shows the smallest improvement, likely due to data distribution mismatch, while data-scaled pre-training, leveraging multiple datasets, provides the most substantial performance enhancement by enabling the model to learn more generalized and robust features.

| Dataset | Backbone | Pretext Tasks | | Validation Set | | |
|---|---|---|---|---|---|---|
| | | Contrastive | Reconstructive | minFDE ↓ | minADE ↓ | MR ↓ |
| | | ✗ | ✗ | 0.969 | 0.661 | 0.092 |
| Argoverse | HiVT | ✔ | ✗ | 0.958 (-1.1%) | 0.656 (-0.8%) | 0.090 (-2.2%) |
| | | ✗ | ✔ | 0.952 (-1.8%) | 0.652 (-1.4%) | 0.090 (-2.2%) |
| | | ✔ | ✔ | **0.940** (-3.0%) | **0.647** (-2.1%) | **0.088** (-4.3%) |

Table 4: Ablations on the two pretext tasks: each task can effectively improve prediction accuracy in isolation, while combining both tasks yields the largest improvement.

datasets. We then analyze these pre-training settings by applying them to HiVT and QCNet, and fine-tune/evaluate them on their original downstream dataset.

The results, presented in Table 3, indicate that: (1) all pre-training strategies effectively improve prediction accuracy compared to training from scratch, demonstrating the effectiveness of utilizing pre-training to learn transferable features; (2) transfer pre-training yields the smallest performance improvement, likely due to the distribution mismatch between the pre-training and downstream datasets, resulting in relatively less relevant learned representations; and (3) data-scaled pre-training, which leverages multiple datasets, provides the most substantial performance enhancement, as the increased diversity in training data helps the model learn more generalized and robust features that benefit the downstream task.

To further demonstrate the effectiveness of utilizing pre-training datasets of our SmartPretrain, we also conduct an ablation on how to use the additional datasets in the pre-training phase. We explored two pre-training settings: 1) We pre-train with the standard motion prediction task on additional datasets and then finetuned on the downstream target dataset. 2) we directly train the downstream target dataset and additional datasets with the standard motion prediction task. The results and discussion are provided in Appendix A.1 due to limited space.

**Ablations on Pretext Tasks.** Recall that our SSL pre-training framework consists of two pretext tasks: trajectory contrastive learning (TCL) and trajectory reconstruction learning (TRL). Table 4 presents an ablation study on them, where we apply them to HiVT on the Argoverse dataset. We observe that each pretext task, when applied in isolation, improves performance—reducing minFDE by 1.1% for TCL and 1.8% for TRL. However, combining both tasks yields the largest improvement, reducing minFDE by 3%.

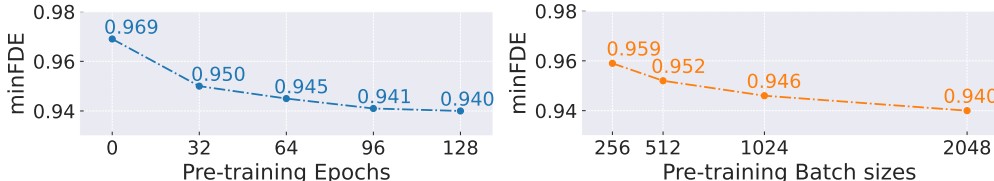

Figure 3: Ablation study on pre-training epochs and batch sizes. Larger pre-training epochs and batch sizes enhance performance, while diminishing returns are observed beyond certain levels.

| Backbone | Category | Reconstruction Target | minFDE ↓ | minADE ↓ | MR ↓ |
|---|---|---|---|---|---|
| | No reconstruction | None | 0.958 | 0.656 | 0.090 |
| HiVT | Reconstruction with historical information | Trajectory of the input sub-scenario | 0.956 (-0.2%) | 0.655 (-0.2%) | 0.090 (-0.0%) |
| | | Trajectory of the entire scenario | 0.948 (-1.0%) | 0.652 (-0.6%) | 0.089 (-1.1%) |
| | Reconstruction with predictive information | Complementary trajectory of the input sub-scenario | **0.940** (-1.9%) | 0.648 (-1.2%) | **0.087** (-3.3%) |
| | | Trajectory of the other sub-scenario | **0.940** (-1.9%) | **0.647** (-1.4%) | 0.088 (-2.2%) |

Table 5: Ablation study on different reconstruction targets. Pre-training with reconstruction tasks boosts prediction performance, with predictive targets showing greater improvement.

**Ablations on Pre-training Epochs and Batch Sizes.** Fig. 3 shows the ablation studies on HiVT and the Argoverse dataset. We observe that larger pre-training epochs and batch sizes improve performance. Larger epochs allow the model to undergo extended training, which helps it learn more nuanced and complex feature representations, ultimately leading to better performance. Meanwhile, larger batch sizes are particularly beneficial for contrastive learning, as they provide a greater number of negative samples within each training batch. However, a diminishing return is observed as we further increase the epochs and batch size, indicating a gap between pretext tasks and the fine-tuning task, and suggesting that beyond a certain point, additional pre-training does not significantly enhance model performance and should be balanced with efficiency to avoid unnecessary overhead.

**Reconstrction Strategies.** Our pre-training pipeline reconstructs the trajectories of the other sub-scenario $t'$ given the input sub-scenario $t$. For the ablation study, we explore more reconstruction targets, categorized into two groups. The first group incorporates predicting historical information, including 1) the trajectory of the input sub-scenario $t$ and 2) the entire scenario. The second group focuses on predictive reconstruction, excluding any historical data, and includes 3) the complementary trajectory steps of the input sub-scenario $t$ and 4) the trajectory of the other sub-scenario $t'$. As shown in Table 5, pre-training with reconstruction tasks enhances prediction performance, with predictive targets yielding the most significant improvements.

**Visulization.** We present some visualization results of fine-tuning in Appendix A.2, categorized into four distinct scenarios: 1) trajectory alignment, 2) long trajectory prediction, 3) novel behavior generation, and 4) smooth and safe trajectory synthesis. These results demonstrate the model's multi-modal trajectory predictions across various scenarios and highlight the enhanced generalization and robustness of the proposed method. We also show some reconstructed trajectories of pre-training in Appendix A.3 to illustrate the effectiveness of our pre-text task learning.

## 5 CONCLUSION

In this paper, we introduce *SmartPretrain*, a novel, model-agnostic, and dataset-agnostic self-supervised learning (SSL) framework designed to enhance motion prediction in autonomous driving. Through a combination of contrastive and reconstructive SSL techniques, SmartPretrain consistently improves the performance of state-of-the-art models across multiple datasets. Our extensive experiments demonstrate significant improvements when applying SmartPretrain to various prediction models like HiVT, HPNet, Forecast-MAE, and QCNet. Additionally, the flexibility of our framework allows effective pre-training across multiple datasets, leveraging data diversity to improve accuracy and generalization. SmartPretrain outperforms existing methods, confirming the effectiveness of our approach. These results highlight SmartPretrain's scalability, versatility, and potential to advance motion prediction in driving environments.

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

# A    APPENDIX

## A.1    ABLATION ON HOW TO PRE-TRAIN ADDITIONAL DATASETS

We sincerely thank Reviewer TuBk for this contributive suggestion on our paper. The original discussion can be seen at: https://openreview.net/forum?id=Bmzv2Gch9v&noteId=xCpSNXeAyo.

We use Argoverse 2 as additional data to pre-train HiVT and Argoverse as the downstream target dataset, we explored two pre-training settings:

- We pre-train the model on Argoverse 2 with the standard motion prediction task, and then fine-tune it to Argoverse.
- We directly train the model on Argoverse and Argoverse 2 with the standard motion prediction task.

A minor design choice is that Argoverse 2 has a longer trajectory horizon than Argoverse. When pre-training on Argoverse 2, we could either randomly sample trajectory segments from the full trajectory, or use a fixed time window. For more comprehensive exploration, we explored both. For the fixed time window choice, considering Argoverse 2 data has 110 waypoints and Argoverse requires 50 waypoints (20 as inputs and 30 as outputs), we use Argoverse 2's original current timestep, and collect 20 historic waypoints as input and 30 future waypoints as output.

We show the results from the first approach (random window) in Table 6 and the results from the second approach (fixed window) in Table 7.

| Backbone | Pre-training Dataset | Finetuning Dataset | Validation Set: Argoverse | | |
|---|---|---|---|---|---|
| | | | minFDE ↓ | minADE ↓ | MR ↓ |
| HiVT | Argoverse 2 | Argoverse | 0.969 | 0.661 | 0.092 |
| | | Argoverse | 1.077 | 0.701 | 0.112 |
| | | Argoverse + Argoverse 2 | 3.359 | 2.092 | 0.636 |

Table 6: Ablation study on motion prediction as the pre-text task for pre-training additional datasets (Argoverse 2 random window). Pre-training with the motion prediction task and then finetuning or directly training with mixed data does not boost prediction performance. They achieve worse prediction results than training from a single dataset.

| Backbone | Pre-training Dataset | Finetuning Dataset | Validation Set: Argoverse | | |
|---|---|---|---|---|---|
| | | | minFDE ↓ | minADE ↓ | MR ↓ |
| HiVT | Argoverse 2 (fixed) | Argoverse | 0.969 | 0.661 | 0.092 |
| | | Argoverse | 1.078 | 0.697 | 0.112 |
| | | Argoverse + Argoverse 2 (fixed) | 1.214 | 0.762 | 0.133 |

Table 7: Ablation study on motion prediction as the pre-text task for pre-training additional datasets (Argoverse 2 fixed window). Pre-training with the motion prediction task and then finetuning or directly training with mixed data does not boost prediction performance. They achieve worse prediction results than training from a single dataset.

Interestingly, for both approaches, poor performance is observed when we directly use motion prediction as the pretraining task, or directly train the model from a mix of the two datasets (especially when we random sample from the additional dataset). It could be presumably due to: 1) the features learned by motion prediction are less transferable or robust, compared to the features learned from SSL tasks; 2) the trajectory distribution between different datasets is quite different, and could be pronounced when pre-training is performed on the standard prediction task.

## A.2 ADDITIONAL VISUALIZATION RESULTS

We present additional visualization results to demonstrate the effectiveness of our method. These results are categorized into four distinct scenarios: (1) trajectory alignment, (2) long trajectory prediction, (3) novel behavior generation, and (4) smooth and safe trajectory synthesis.

**Trajectory Alignment.** As illustrated in Fig. 4, we showcase scenarios involving the prediction error of the target agent's trajectories is consistently reduced after pre-training. With more datasets used in pre-training, the trajectories get closer to the ground truth, showing enhanced generalization and robustness with multiple datasets pre-training.

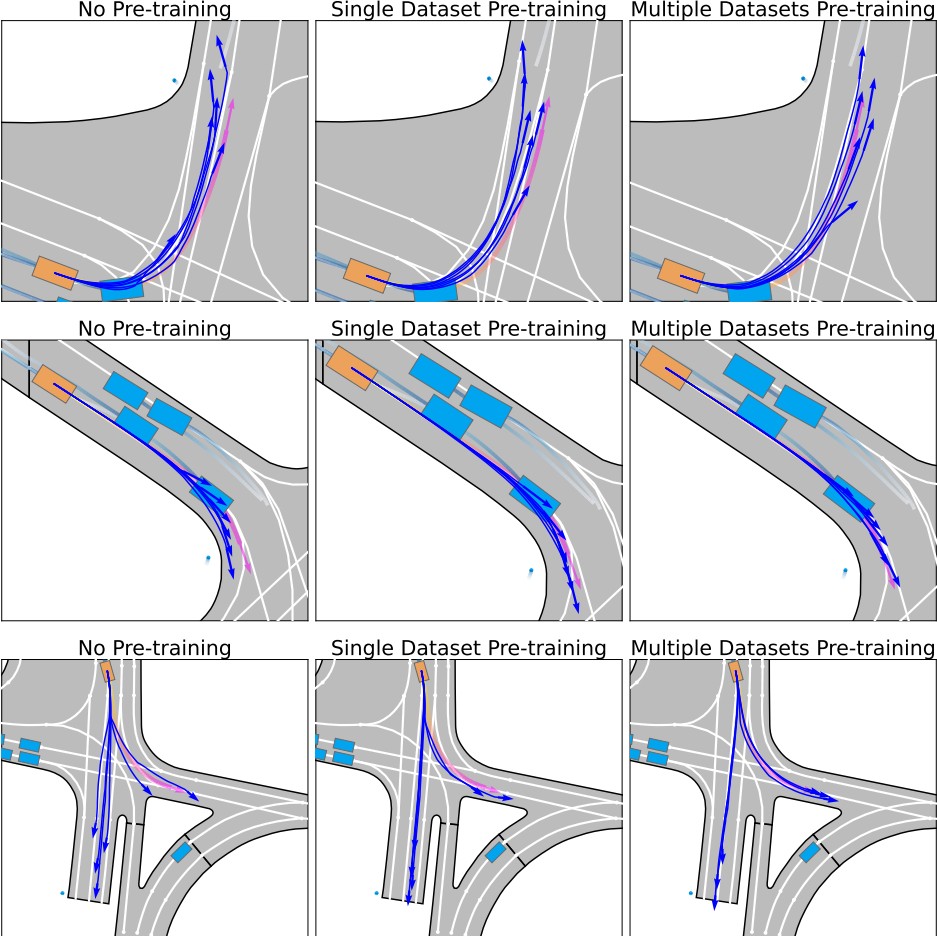

Figure 4: Visulization Results of Trajectory Alignment. The blue arrows are the model's multi-modal trajectory predictions for the target agent, and the pink arrow is the ground truth future trajectory. After pre-training, the predicted trajectories get closer to the ground truth.

**Long Trajectories.** As illustrated in Fig. 5, we showcase scenarios involving long trajectory predictions compared against ground truth data. Incorporating additional datasets during pre-training enhances the model's ability to capture fine-grained spatiotemporal interactions with objects within the scenario, thereby enabling more accurate trajectory predictions in the long term.

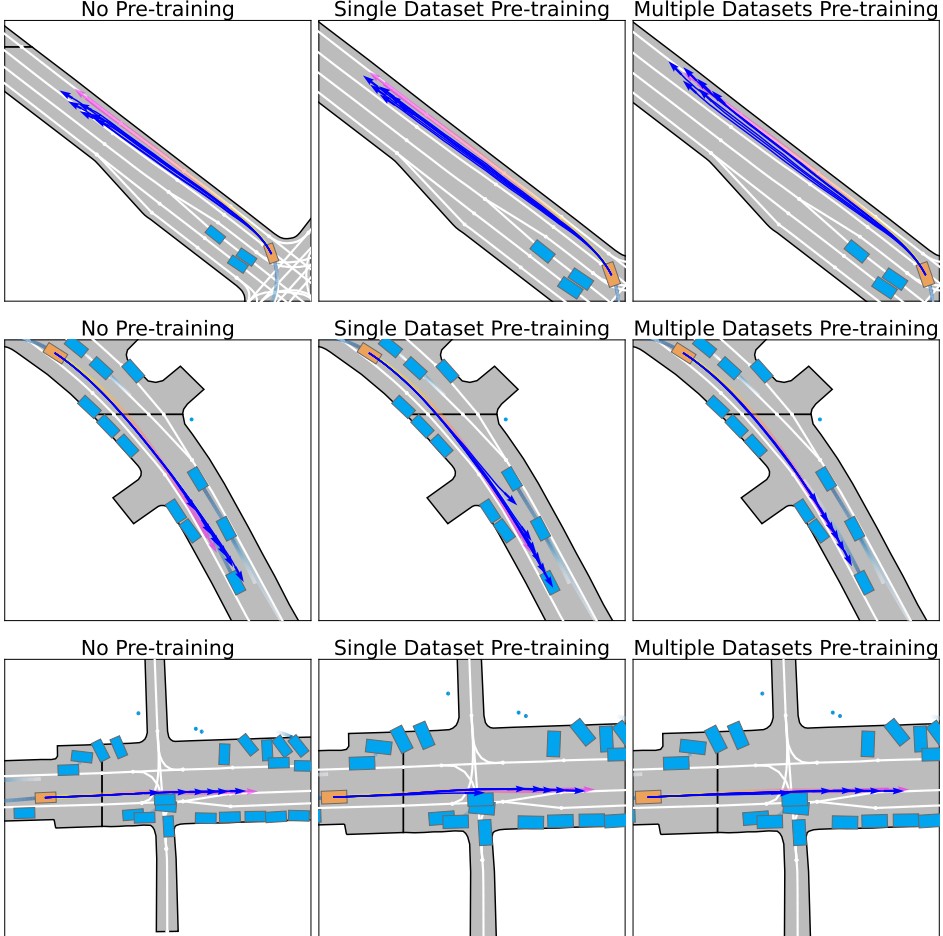

Figure 5: Visulization Results of Long Trajectories. The blue arrows are the model's multi-modal trajectory predictions for the target agent, and the pink arrow is the ground truth future trajectory. Our pre-training method enables more accurate trajectory predictions in the long term.

**Behavior Generation.** We present some scenarios in which new behavior is learned through pre-training. In Fig. 6, the original model struggles to align well with the ground truth across six modality outputs. However, after pre-training, particularly multiple datasets pre-training, the model acquires new behaviors, resulting in the generation of more meaningful and skillful trajectories. For example, lane-changing maneuvers (Fig. 6 top), lane-keeping (Fig. 6 middle) and trajectories that are closely synchronized with the ground truth (Fig. 6 bottom).

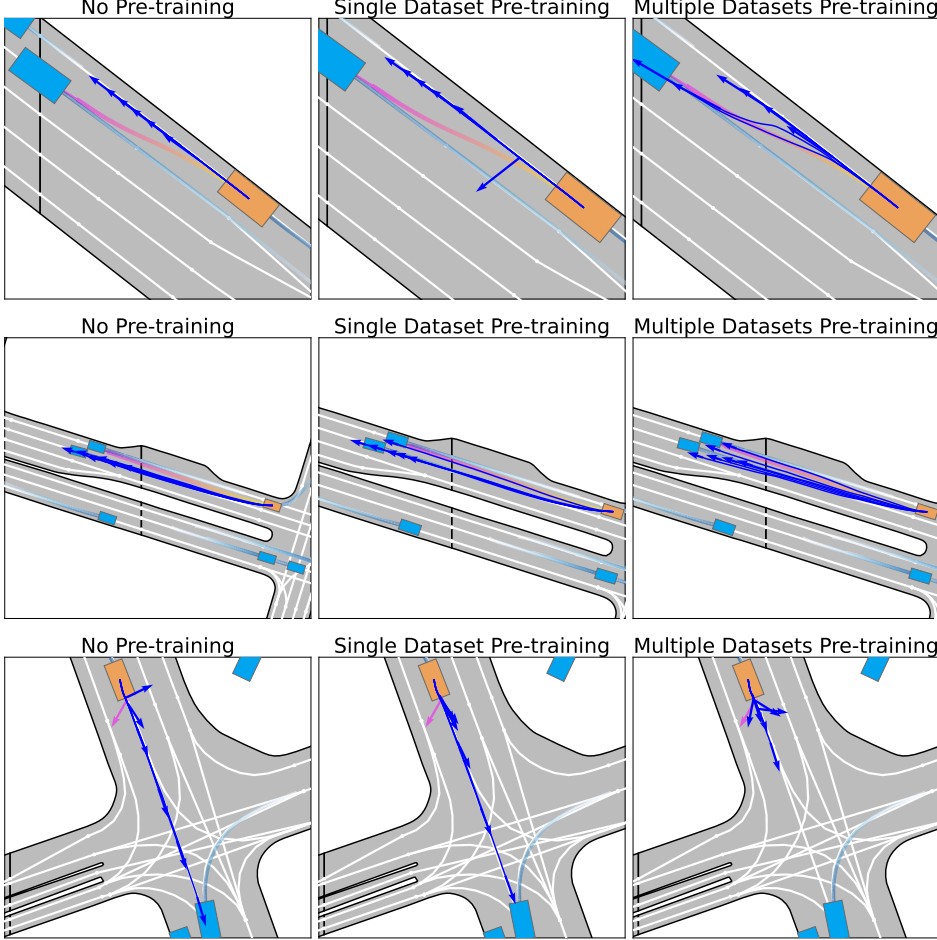

Figure 6: Visulization Results of Behavior generation. The blue arrows are the model's multi-modal trajectory predictions for the target agent, and the pink arrow is the ground truth future trajectory. Our pre-training method can teach new behavior learned through multiple datasets to the model, thus generating more meaningful and skillful trajectories.

**Smoothness and Safety Synthesis.** In Fig. 7, we illustrate scenarios where pre-training facilitates the generation of smoother and safer trajectories compared to models without pre-training. Coarse trajectories are improved into smoother and safer ones, demonstrating the generalizability and robustness of the learned features achieved through our multiple datasets pre-training.

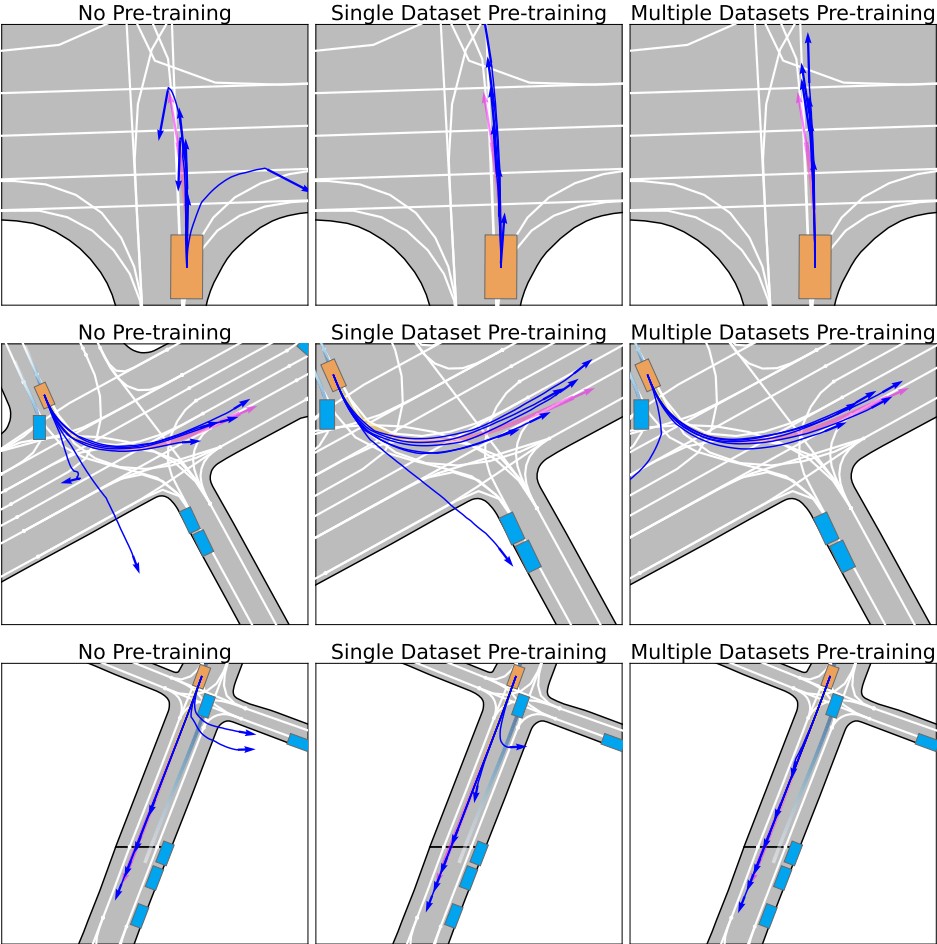

Figure 7: Visulization Results of Smooth and Safety Synthesis. The blue arrows are the model's multi-modal trajectory predictions for the target agent, and the pink arrow is the ground truth future trajectory. Our pre-training method contributes to learning smoother and safer trajectories, especially through multiple datasets pre-training.

## A.3 RECONSTRUCTED TRAJECTORIES

We present visualization results of reconstructed trajectories to demonstrate the effectiveness of our pre-text task learning. As shown in Fig. 8, the reconstructed trajectories align closely with the ground truth, indicating the successful learning of our pretext task.

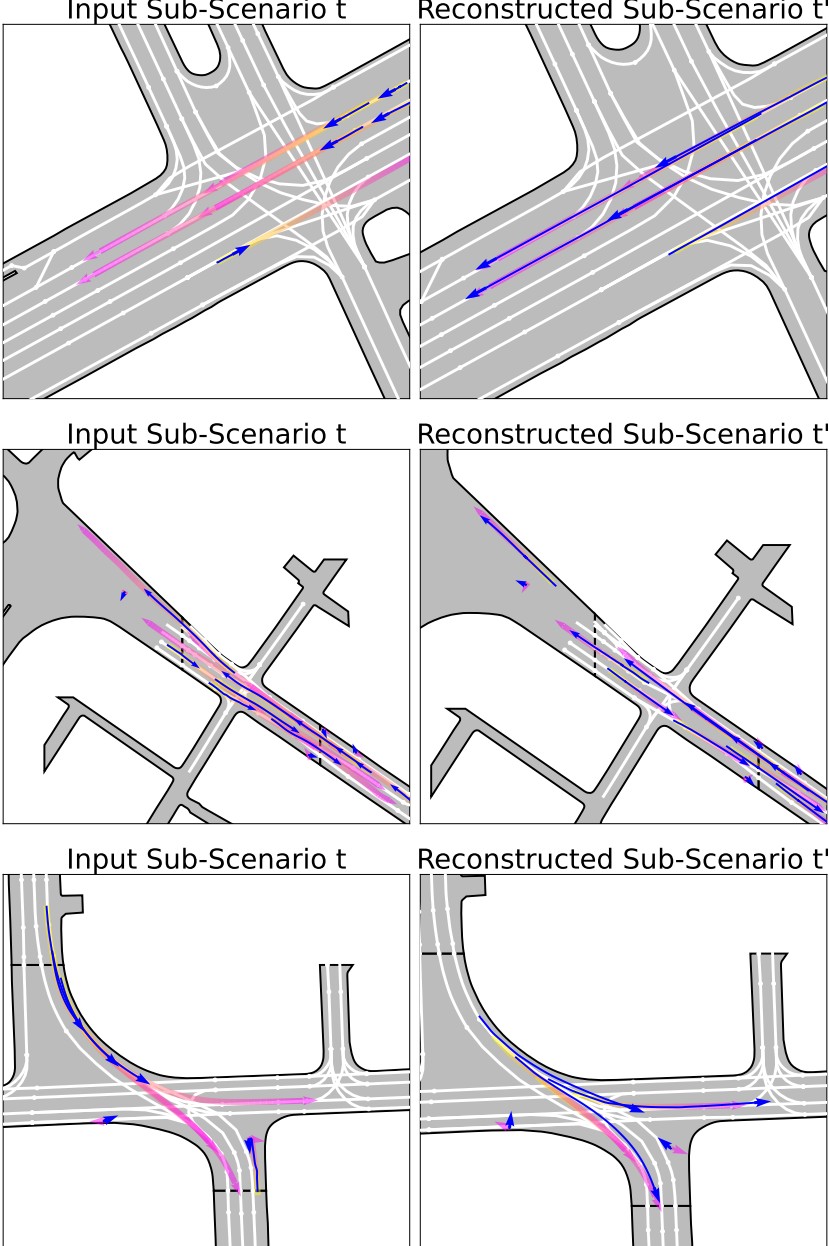

Figure 8: Visualization Results of Reconstructed Trajectories. **Left:** The blue arrows are the trajectories of sub-scenario $t$, and the pink arrows are the trajectories of the entire scenario. **Right:** The blue arrows are the reconstructed trajectories of sub-scenario $t'$, and the pink arrows are the ground truth trajectories of the sub-scenario $t'$. Through pre-training, the model successfully reconstructs the sub-scenario $t'$ based on sub-scenario $t$. The promising results suggest that the reconstruction task has been effectively learned.

