# OpenReview forum: "SmartPretrain: Model-Agnostic and Dataset-Agnostic Representation Learning for Motion Prediction"
_ICLR.cc/2025/Conference — ICLR 2025 Poster_

### Official Review · Reviewer_TuBk · 2024-10-28

**Soundness:** 3
**Presentation:** 3
**Contribution:** 2
**Rating:** 8
**Confidence:** 5

**Summary:**

This paper offers a universal pipeline to pre-train on real-world motion data for trajectory prediction tasks. Combining popular self-supervised pre-training methods like contrastive learning and reconstruction learning, and with specific designs on motion data domain, this work successfully proposes a pre-training pipeline to learn general representations of *trajectories* in motion data, which can work regardless of the baseline motion prediction models or the motion datasets used. Extensive experiments have been performed on various commonly-used datasets (Argoverse 1/2, Waymo Open Motion Dataset) and state-of-the-art baselines (HiVT/QCNet etc.) to show the effectiveness of the pre-training pipeline proposed. A series of ablation studies clearly ablate the effectiveness of each module of the pre-training pipeline as well as the pre-training data involved.

**Strengths:**

1. The pipeline proposed in this work is model-agnostic, i.e., it can be easily extended to any encoder-decoder style motion prediction model.
2. This work successfully combines different real-world datasets like Argoverse and Waymo, so it can significantly enlarge the available motion data that can be used for a specific motion prediction task, where data scarcity is a significant problem.
3. Extensive experiments are conducted to prove its general effectiveness across different baseline models and datasets.
4. Ablation studies are well designed to clearly show i) the effectiveness of each part of the pipeline ii) the influence of pre-training data.

**Weaknesses:**

1. The downstream motion prediction settings, though already very diverse, seem not being able to cover all necessary cases. For example, no methods fine-tuned on Waymo Open Motion Dataset (WOMD) are presented.
2. The pre-training performance should be illustrated to prove that pre-training tasks can be done successfully. For example, can you show some examples of reconstructed trajectories?
3. Direct data mixing in Data-scaled Pre-training is a natural choice, but might not be optimal. For example, WOMD has significant domain gap compared to Argoverse. In this case, a biased weight might be helpful in pre-training stage to lower the influence of WOMD-Argoverse domain gaps.
4.  An ablation on how to utilize the additional data in the pre-training stage could be added to make Table 3 even more convincing. For example, in Transfer Pre-training and Data-scaled Pre-training, what would happen if the additional data is used to pre-train on the baseline model directly, or even directly to augment the training set for the baseline model?
5. Some minor questions:
i) Why to use L1 loss in TRL instead of L2, while the latter is the base for metrics used (MR, FDE, ADE)?
ii) The quantity of data that is complete / incomplete might be presented to help readers understand the quantity of additional data introduced through this work.
6. The prediction metrics lack error bars. This is not a weakness, but a point that can be even improved. I understand that the motion prediction metrics can be unstable sometimes, but adding error bars onto the most important results would significantly improve the reliabilities of the pipeline proposed.

**Questions:**

Please see the Weaknesses. I am looking forward to the authors' rebuttal and discussions on those issues.

---

> ### Author Response · Authors · 2024-11-21
> **Response to Reviewer TuBk (1/3)**
>
> Dear reviewer `TuBk`,  we sincerely appreciate the thorough assessment and contributive suggestions on our paper! We address each of your questions as follows.
>
> > The downstream motion prediction settings, though already very diverse, seem not being able to cover all necessary cases. For example, no methods fine-tuned on Waymo Open Motion Dataset (WOMD) are presented.
>
> Thanks for pointing this out! In this paper, we present an early attempt to scale multiple trajectory datasets for pre-training in motion prediction. However, training with scaled data has proven to be both GPU-intensive and time-consuming. Upon closer examination, the Argo 1, Argo 2, and WOMD provide 250k, 200k, and 487k training scenarios, respectively. Among these, the WOMD stands out due to its larger size, a greater number of vehicles per data sample, and the need for a more complex data loader. Additionally, models designed for WOMD (e.g., MTR) are typically larger and more resource-intensive than those for Argo or Argo 2.
>
> Given our limited computational resources, we focused on Argo and Argo 2 for this paper and did not include results on WOMD at the time of submission. However, we fully agree that fine-tuning on WOMD is crucial for evaluating cross-dataset pre-training in motion prediction. While these experiments are time-consuming to run, we are currently making every effort to obtain these results and will add them to our paper as soon as they become available. Thank you for this excellent suggestion to help make our paper more complete!
>
> > The pre-training performance should be illustrated to prove that pre-training tasks can be done successfully. For example, can you show some examples of reconstructed trajectories?
>
> Thanks for this suggestion. We've added some visualization results of reconstructed trajectories in Appendix A.2 of the revised manuscript. As also suggested by reviewer 2, we've added some more intuitive visualization results of fine-tuning in Appendix A.1.
>
> > Direct data mixing in Data-scaled Pre-training is a natural choice, but might not be optimal. For example, WOMD has significant domain gap compared to Argoverse. In this case, a biased weight might be helpful in pre-training stage to lower the influence of WOMD-Argoverse domain gaps.
>
> Thanks for this insightful comment. Yes, we've done some experiments in our early experiments when exploring data scaling and balancing. Specifically, we use a weight of 40% to WOMD and it results in about 200k data which is about 1:1 with Argo. As shown below, we explored pretraining with WOMD, and pretraining with mixing of the two datasets.
>
>
> | Pre-Trainng Datasets | Fine-Tuning Dataset | minFDE | minADE | MR    |
> | -------------------- | ------------------- | ------ | ------ | ----- |
> | \                    | Argo                | 0.969  | 0.661  | 0.092 |
> | WOMD_0.4             | Argo                | 0.950  | 0.653  | 0.088 |
> | WOMD_1.0             | Argo                | 0.946  | 0.652  | 0.089 |
> | Argo+WOMD_0.4        | Argo                | 0.937  | 0.647  | 0.087 |
> | Argo+WOMD_1.0        | Argo                | 0.935  | 0.645  | 0.086 |
>
> As shown in the table, in both pre-training settings, the fine-tuning performance slightly drops when we only use 0.4 of the WOMD data, presumably due to the increased diversity of the data.

---

> ### Author Response · Authors · 2024-11-21
> **Response to Reviewer TuBk (2/3)**
>
> > An ablation on how to utilize the additional data in the pre-training stage could be added to make Table 3 even more convincing. For example, in Transfer Pre-training and Data-scaled Pre-training, what would happen if the additional data is used to pre-train on the baseline model directly, or even directly to augment the training set for the baseline model?
>
> Thanks for sharing this interesting idea. As suggested, using Argo2 as additional data to pre-train HiVT and Argo as the downstream target dataset, we explored two pre-training settings:
>
> 1. We pre-train the model on Argo 2 with the standard motion prediction task, and then fine-tune it to Argo.
> 2. We directly train the model on Argo and Argo 2 with the standard motion prediction task.
>
> A minor design choice is that, Argo 2 has a longer trajectory horizon than Argo. When pre-training on Argo 2, we could either randomly sample trajectory segments from the full trajectory, or use a fixed time window. For more comprehensive exploration, we explored both. For the fixed time window choice, considering Argo2 data has 110 waypoints and Argo1 requires 50 waypoints (20 as inputs and 30 as outputs), we use Argo2’s original current timestep, and collect 20 historic waypoints as input and 30 future waypoints as output.
>
>
> We show the results from the first approach (random window) in the table below.
> | Pre-Training Dataset | Fine-Tuning Dataset | minFDE | minADE | MR    |
> | -------------------- | ------------------- | ------ | ------ | ----- |
> | \                    | Argo                | 0.969  | 0.661  | 0.092 |
> | Argo2                | Argo                | 1.077  | 0.701  | 0.112 |
> | \                    | Argo+Argo2          | 3.359  | 2.092  | 0.636 |
>
> We also show the results from the second approach (fixed window) in the table below.
> | Pre-Training Dataset | Fine-Tuning Dataset | minFDE | minADE | MR    |
> | -------------------- | ------------------- | ------ | ------ | ----- |
> | \                    | Argo                | 0.969  | 0.661  | 0.092 |
> | Argo2 (fixed)        | Argo                | 1.078  | 0.697  | 0.112 |
> | \                    | Argo+Argo2 (fixed)  | 1.214  | 0.762  | 0.133 |
>
> Interestingly, for both approaches, poor performance is observed when we directly use motion prediction as the pretraining task, or directly train the model from a mix of the two datasets (especially when we random sample from the additional dataset). It could be presumably due to: 1) the features learned by motion prediction are less transferable or robust, compared to the features learned from SSL tasks; 2) the trajectory distribution between different datasets is quite different, and could be pronounced when pre-training is performed on the standard prediction task.
>
> We believe these interesting findings will contribute valuable insights to the field. We sincerely thank the reviewer for this constructive idea, and we will incorporate these results into Table 3 and re-organize the corresponding section in our final paper to present the findings more clearly and cohesively.

---

> ### Author Response · Authors · 2024-11-21
> **Response to Reviewer TuBk (3/3)**
>
> > Some minor questions: i) Why to use L1 loss in TRL instead of L2, while the latter is the base for metrics used (MR, FDE, ADE)? ii) The quantity of data that is complete / incomplete might be presented to help readers understand the quantity of additional data introduced through this work.
>
> Thanks for the suggestion. As common in the literature, we consider L1 loss instead of L2 loss due to practical considerations related to:
>
> 1. Robustness to outliers: L2 loss penalizes large errors more heavily due to the squaring term, making it sensitive to outliers and noises, while L1 loss could be relatively more robust.
> 2. Balanced training signal: In the early stage of training, where the reconstruction errors could be high and stochastic, L1 loss could provide a more stable learning signal. In the later stage of training, where the reconstruction errors are usually small, L1 loss could provide a stronger learning signal, since the square operation in L2 loss would further reduce its loss value.
>
> During rebuttal, we also conducted one experiment using L2 loss during pre-training, to study their differences. As shown in the table below, L2 reconstruction results in less improvement compared with L1 reconstruction.
>
> | Pre-Training Setup | minFDE | minADE | MR    |
> | ------------------ | ------ | ------ | ----- |
> | None               | 0.969  | 0.661  | 0.092 |
> | L1 reconstruction  | 0.940  | 0.647  | 0.088 |
> | L2 Reconstruction  | 0.948  | 0.654  | 0.089 |
>
> As for the number of complete data in these datasets, sure, the percentage of after-filtering complete trajectories, over all vehicle trajectories is 35%, 27% and 25% for Argo, Argo2 and WOMD respectively.
>
> > The prediction metrics lack error bars. This is not a weakness, but a point that can be even improved. I understand that the motion prediction metrics can be unstable sometimes, but adding error bars onto the most important results would significantly improve the reliabilities of the pipeline proposed.
>
> Thanks for pointing out this. Indeed, most motion prediction methods in the literature have no error bars as the training cost is relatively high. For example, the training of QCNet and HPNet takes about 2 days. We sincerely value your proposal and agree that adding error bars will improve the reliabilities of our proposed pipeline. So we chose a fast training setup (HiVT pre-training) and repeated our pipeline's pre-training and fine-tuning three times with different random seeds. The results are listed in the following table.
>
> | Random Seed | minFDE         | minADE         | MR          |
> | ----------- | -------------- | -------------- | ----------- |
> | 2023        | 0.939          | 0.647          | 0.088       |
> | 2024        | 0.940          | 0.649          | 0.088       |
> | 2025        | 0.939          | 0.646          | 0.088       |
> | mean / std  | 0.939 / 5.7e-4 | 0.646 / 1.5e-3 | 0.088 / 0.0 |
>
> The performance is relatively stable, demonstrating that our proposed pipeline learns robust features for fine-tuning.

---

> > ### Comment · Reviewer_TuBk · 2024-11-22
> > **Great job!**
> >
> > I would like to thanks the authors for providing detailed experiments and explanations on all my questions and concerns. The rebuttals are convincing to me. I would raise my evaluation to accept to recognize the improved reliability of the proposed method, considering the evidences in the rebuttal experiments.
> >
> > I would sincerely hope that the authors could re-organize these additional experiments into the manuscript, during the camera-ready phase if accepted. I believe these would further help to convince readers on the effectiveness of the pre-training. And again, I highly appreciate the author's thorough and solid evidences provided.

---

> > > ### Author Response · Authors · 2024-11-25
> > >
> > > Dear Reviewer `TuBk`,
> > >
> > > Thank you for your feedback and for raising your score to 8 accept! We really appreciate your contributive suggestions to significantly improve our work. We fully agree that these additional results would further help to demonstrate the effectiveness of the pre-training, and further promote more explorations toward scaling law in motion prediction community. We will certainly emphasize and organize these points in the final version. Thank you once again for your valuable insights and recommendations!

---

### Official Review · Reviewer_ND2Q · 2024-11-01

**Soundness:** 3
**Presentation:** 3
**Contribution:** 3
**Rating:** 8
**Confidence:** 4

**Summary:**

This paper introduces a self-supervised learning framework for motion forecasting that is both model-agnostic and dataset-agnostic. The approach unifies data samples from various motion forecasting datasets, such as WOMD, AV1, and AV2, making it feasible to pretrain models on large-scale, multi-source data. The framework incorporates both conservative learning and a reconstruction task, achieved through Trajectory Contrastive Learning (TCL) and Trajectory Reconstruction Learning (TRL). Experimental results demonstrate significant performance improvements across multiple architectures and datasets, validating the effectiveness of the proposed methods.

**Strengths:**

1. This paper is well-organized and easy to follow. The experimental results are thorough, covering various datasets and methods, and providing strong evidence for the method's effectiveness.

2. While conservative learning and reconstruction tasks are common in self-supervised learning frameworks for motion forecasting, this submission introduces some innovative strategies that add value.

3. The approach to unifying data representation from diverse sources could pave the way toward a foundation model for motion forecasting.

**Weaknesses:**

While the novelty of this submission may be somewhat limited and most techniques are already verified in many previous works, it does not present any clear weaknesses. For specific considerations, please refer to the questions section.

**Questions:**

1. In the experiment section (Table 1), it is noted that both HPNet and Forecast-MAE were not pretrained on all three datasets, reportedly due to "compute constraints." This reasoning should be clarified further to help readers understand the specific limitations or challenges involved.

2. The reconstruction task is commonly employed in motion forecasting pretraining frameworks, with two main approaches: predicting masked tokens (as in Forecast-MAE) or predicting masked tail trajectories (as in SEPT[2]). The proposed method follows a strategy similar to the latter, which has been shown to outperform the token prediction approach in [2]. It would be beneficial to emphasize the main distinctions of the proposed method from this established approach to further highlight its contributions.

[1] Forecast-MAE: Self-supervised Pre-training for Motion Forecasting with Masked Autoencoders

[2] SEPT: Towards Efficient Scene Representation Learning for Motion Prediction

---

> ### Author Response · Authors · 2024-11-21
> **Response to Reviewer ND2Q**
>
> Dear reviewer `ND2Q`, we sincerely appreciate the thoughtful review and precious feedback on our paper! We have carefully addressed your concerns as outlined below.
>
> > In the experiment section (Table 1), it is noted that both HPNet and Forecast-MAE were not pretrained on all three datasets, reportedly due to "compute constraints." This reasoning should be clarified further to help readers understand the specific limitations or challenges involved.
>
> Thanks for pointing this out! In this paper, we present an early attempt to scale multiple trajectory datasets for pre-training in motion prediction, drawing inspiration from the successes in NLP and CV. However, training with scaled data has proven to be both GPU-intensive and time-consuming. Furthermore, exploring the influence of different dataset combinations could easily double the number of required experiments, significantly increasing the computational cost.
>
> Given these constraints and the limited time before the paper submission, we focused our experiments on data-scaled pre-training with one model per dataset, ultimately selecting HiVT for Argo and QCNet for Argo2. In Table 3, we present the effects of different dataset scaling strategies, such as same-dataset pre-training, cross-dataset pre-training, and pre-training on all datasets. We believe these experiments provide valuable signals and insights to the community. Additionally, as recommended by the reviewer `TuBk`, we are now exploring more pre-training settings on WOMD, which we believe will further enhance the completeness and potential impact of the proposed method.
>
> > The reconstruction task is commonly employed in motion forecasting pretraining frameworks, with two main approaches: predicting masked tokens (as in Forecast-MAE) or predicting masked tail trajectories (as in SEPT[2]). The proposed method follows a strategy similar to the latter, which has been shown to outperform the token prediction approach in [2]. It would be beneficial to emphasize the main distinctions of the proposed method from this established approach to further highlight its contributions.
>
> Thanks for pointing out this and it's a valuable question! The main distinction of our reconstruction task from SEPT and Forecast-MAE is that our input and reconstruction target are not fixed, and are temporally varied depending on the subs-scenario sampling (random t and t’). Therefore our reconstruction task is more challenging due to the more randomness introduced to input and output trajectories. It contributes to learning more informative and transferable features.
>
> Besides, we will open-source our code upon acceptance of our paper to further contribute to the community, and here we provide a rough and initial preview for our code and checkpoint in this anonymous [URL](https://anonymous.4open.science/r/5f404fda8de3e3278e2f794f80bffed0036c827b).

---

> > ### Comment · Reviewer_ND2Q · 2024-11-27
> >
> > Thank you for your clarification. Please incorporate these changes into the final version to enhance the paper's clarity. I will update my score to 8.

---

> > > ### Author Response · Authors · 2024-11-27
> > >
> > > Dear Reviewer `ND2Q`,
> > >
> > > Thank you for your feedback and for raising your score to 8 accept! We really appreciate your precious suggestions to significantly improve our work. We will certainly emphasize and incorporate these changes in the final version. Thank you once again for your valuable insights and recommendations!

---

### Official Review · Reviewer_wPEn · 2024-11-03

**Soundness:** 3
**Presentation:** 3
**Contribution:** 2
**Rating:** 5
**Confidence:** 4

**Summary:**

In this paper, the authors present SmartPretrain, a novel self-supervised learning (SSL) framework that is model-agnostic and dataset-agnostic. This framework aims to overcome the challenges associated with the scarcity of large-scale driving datasets for motion prediction and the reliance of existing SSL pre-training methods on specific model structures. SmartPretrain incorporates both contrastive and reconstructive SSL approaches and features a dataset-agnostic scenario sampling strategy that combines multiple datasets. Extensive experiments validate the effectiveness of SmartPretrain in motion prediction.

**Strengths:**

1.The field is worthy researching, and the motivations behind the method is clear.
2.The paper is well-organized.
3.The paper proposes to pre-train from composition of different sources data, which is rarely researched in motion prediction areas before.

**Weaknesses:**

1. While the method introduces a novel paradigm for trajectory prediction, the pretrain-finetune approach has been widely adopted across various domains like NLP and CV for years, making it less valuable. Consequently, the contribution of SSL for model training is incremental.
2. The technical contribution of the dataset sampling strategy appears limited: aspects like standardizing representations, ensuring data quality, and maximizing volume and diversity are fundamental considerations when integrating different data sources.
3. With only one SSL pre-training baseline and a single dataset in Table 2, it may be challenging to substantiate the proposed method’s advantages over other SSL approaches.
4. If possible, additional visualization results from the authors would be highly valuable.
5. The pre-training needs 32 Nvidia A100 40GB GPUs for 128 epochs, which takes abundant computational resources. However, the improvements are not that significant.
In short, the contributions of the papers are limited, especially the technical part. I think the paper cannot meet the standard of ICLR conference,

**Questions:**

Please see weaknesses.

---

> ### Author Response · Authors · 2024-11-21
> **Response to Reviewer wPEn (1/2)**
>
> Dear reviewer `wPEn`, we sincerely appreciate the careful review and valuable feedback on our paper!  We have addressed each of your concerns as follows.
>
> > While the method introduces a novel paradigm for trajectory prediction, the pretrain-finetune approach has been widely adopted across various domains like NLP and CV for years, making it less valuable. Consequently, the contribution of SSL for model training is incremental.
>
> Thank you for highlighting this concern. We completely agree that the pretrain-finetune approach has been widely adopted in NLP and CV for years, with numerous renowned works establishing its effectiveness and significance. However, this approach has been much less explored in the motion prediction domain, which presents unique challenges compared to the NLP and CV domains. Unlike the relatively uniform data formats in CV (images) and NLP (tokens), where pixels and text provide straightforward representations, motion prediction relies on diverse and multi-modal data sources such as maps and motion trajectories. Map representations alone exhibit significant variability (e.g., rasterized maps versus vectorized maps), and different datasets often use distinct formats for motion data, further compounding the complexity of this domain. This requires specific domain knowledge of motion prediction tasks to design effective SSL techniques, as demonstrated by recent efforts such as SEPT (ICLR'24) and Forecast-MAE (ICCV'23).
>
> In this context, our SSL task design incorporates a novel contrastive learning objective, which aligns the same agent's embeddings across different time windows. While this approach has not been introduced in previous motion prediction methods, we want to note that our major focus and primary contribution lie in enabling general pre-training across models and datasets for motion prediction, which has not been achieved by prior works. Our main goal is to introduce the first general SSL framework that can be universally applied to various motion prediction models, which is why we designed our pretext tasks in an agent-centric manner. Furthermore, we are among the first to perform data-scaled pretrain-finetune for motion prediction. Through model-agnosticism and dataset-agnosticism, we aim to present an early exploration of the 'scaling laws' in the motion prediction domain, an area that has been significantly underexplored.
>
> Besides, the code will be open-sourced, and we here provide an initial and rough preview of it through this anonymous [URL](https://anonymous.4open.science/r/5f404fda8de3e3278e2f794f80bffed0036c827b) for preview. By making our implementation publicly available, we aim to foster transparency and reproducibility, and provide a foundation for further research and development in the motion prediction domain.
>
> > The technical contribution of the dataset sampling strategy appears limited: aspects like standardizing representations, ensuring data quality, and maximizing volume and diversity are fundamental considerations when integrating different data sources.
>
> Thanks for the question. First, we fully agree that standardizing representations, ensuring data quality, and maximizing volume and diversity are essential practices for integrating different data sources. However, similar as mentioned in our previous response, unlike well-established domains such as NLP and CV, motion prediction presents unique challenges that require specific domain knowledge to address effectively. For example, while the CV community has well-established and straightforward data-mixing practices for standardizing representations (e.g., lighting and coloring), ensuring data quality (e.g., removing corrupted images), and diversifying data distributions (e.g., geographic diversity), clear and comprehensive solutions for achieving these goals remain elusive in the motion prediction domain. Due to these complexities, previous SSL approaches in motion prediction (e.g., Forecast-MAE, SEPT, TrajMAE, PreTram, and others) have not ventured into data scaling or mixing. We are among the first to investigate data scaling and mixing in the trajectory domain and propose practical techniques to achieve it. We will also open-source our code to promote transparency and collaboration. With the comprehensive studies of different data combinations, we hope our insight from mixing datasets pre-training inspires more researchers in the community to value data scaling/mixing in pre-training of motion prediction, ultimately leading to more thoughtful and effective utilization of motion prediction datasets.

---

> ### Author Response · Authors · 2024-11-21
> **Response to Reviewer wPEn (2/2)**
>
> > With only one SSL pre-training baseline and a single dataset in Table 2, it may be challenging to substantiate the proposed method’s advantages over other SSL approaches.
>
> Thank you for the question. As mentioned in line 403-404 of our paper, to ensure a fair and meaningful comparison, we focused on SSL methods with open-source code. However, to the best of our knowledge, at the time of this paper’s submission, Forecast-MAE was the only open-sourced SSL method available for Argo and Argo 2. Consequently, we included only one comparison in Table 2, where our method demonstrated a significantly larger improvement over Forecast-MAE’s pre-training method. Beyond the performance boost, we would also like to highlight that, while prior SSL methods lack generality and can only be applied exclusively to a single model/dataset, our SSL pretraining strategy is designed to be flexibly applicable across models and datasets, showing broader applicability.
>
> Besides, we believe that the seemingly limited comparisons with other SSL approaches are not a key weakness of our work, but rather highlight the current gaps and the underexplored nature of this research area, which calls for more contributions from the community. To help address this, we will open-source our work, with the hope of further accelerating progress in this field.
>
> > If possible, additional visualization results from the authors would be highly valuable.
>
> Thanks for this suggestion. Yes, we've added more intuitive visualization results of fine-tuning in Appendix A.1 of the revised paper. As also suggested by reviewer `TuBk`,  we've added some visualization results of reconstructed trajectories of pre-training in Appendix A.2.
>
> > The pre-training needs 32 Nvidia A100 40GB GPUs for 128 epochs, which takes abundant computational resources. However, the improvements are not that significant. In short, the contributions of the papers are limited, especially the technical part. I think the paper cannot meet the standard of ICLR conference,
>
> Thanks for this careful review. We've realized the training cost descriptions have not been comprehensive in our original paper. The need of 32 Nvidia A100 40GB GPUs is only applicable when we conduct pretraining on all datasets together, which is our maximum training cost setting: all datasets will contribute to around 900k data and prolonged training cost, thus we use more GPUs to accelerate training. As for single-dataset pre-training, we use 8 GPUs. We have updated the claim and made it more clear in line 367-369 of our revised paper. As in Fig. 3, pre-training with 32 epochs can already have an effective performance boost compared with no pre-training (minFDE 0.950 v.s. 0.969), and further increasing training epochs leads to diminishing returns. We pre-train 128 epochs just to explore the model limit when we scale up the training compute for motion prediction.
>
> Besides, similar to data scaling in CV and NLP (e.g., ImageNet and the 400B-token datasets for GPT training), the training cost is an inevitable part of exploring scaling laws. To mitigate the need for repeated training, we will share our pre-trained model weights learned from various data sources. We believe our detailed experiments with diverse models and datasets will provide valuable research insights to the community, such as the design of a general pre-training framework for motion prediction and strategies for data mixing in the trajectory domain. With these techniques rarely explored before, and combined with our open-sourced code, we aim to address the "rarely researched and worth researching field", "pave the way toward a foundation model for motion forecasting" and tackle the issue of "data scarcity is a significant problem", as acknowledged by you, Reviewers `ND2Q` and `TuBk`.

---

### Official Review · Reviewer_SCK1 · 2024-11-03

**Soundness:** 3
**Presentation:** 3
**Contribution:** 3
**Rating:** 6
**Confidence:** 4

**Summary:**

The work proposes a pretraining self-supervised learning framework that can be applied to many models, and trained on different datasets for motion prediction. The pretraining pipeline leverages momentum contrast and generates contrastive pairs by augmenting the same traffic scene with non-overlapping time horizon clips, for the contrastive loss of embeddings and trajectory reconstruction loss. It demonstrates better performance with the pretraining pipeline in two different datasets and various models.

**Strengths:**

1. The paper is well-written and easy to follow.
2. Good performance is achieved on the AV, AV2 datasets, with different models.
3. The exploration of the concept of model-agnostic and dataset-agnostic is very good.

**Weaknesses:**

1.	Many details are missing, which may hinder reproducibility. For instance, in the pretraining phase, it is unclear how the values for t and t' are selected for each dataset to avoid overlapping for the experiments. The horizons of the sub-scenario as input and reconstruction are also not specified; it would be helpful to know if these are consistent with the motion prediction settings (either input or output horizons?) used during fine-tuning. Additionally, the default lambda value for the loss function is not provided. Clarifications on which parts of each network are used for pretraining would be beneficial (refer to question 1 below). Will the code be made available as open-source?
2.	Computational cost is not shown and compared.

**Questions:**

1.	In Fig 2, the figure labels the component as "model" for pretraining. However, since Section 3.1 (problem formulation) indicates that the contrastive loss is calculated on the encoded embeddings, should this component be referred to as the "encoder" instead? In the experiments with HiVT, HPNet, QCNet, and Forecast-MAE, are only the encoders of these models pretrained? For models like HPNet, which do not clearly differentiate between encoder and decoder architectures, how do you determine which parts of the model to use for pretraining to obtain latent embeddings? How might this selection influence the results?
2.	line 95-97, I find this claim unclear and potentially misleading. I disagree that MAE pretraining is inflexible; on the contrary, I think the masking pretraining is quite versatile. The masking techniques used in the papers you referenced - Rmp, Traj-mae, Forecast-mae, Sept - appear very similar, suggesting the masking concept is not limited and can be readily applied across various models.
3.	Line 104, abbreviation CL is not explained before.
4.	Line 266 mentions that only complete trajectories are used in pretraining, excluding incomplete ones. During pretraining phase, do you reconstruct single-agent trajectory or multi-agent? The same question applies to the prediction phase. Could you specify what percentage of trajectories remain after filtering for each dataset?
5.	Line 296, for eq.(1), why the case of i=j is not excluded in the second term of the denominator, as in this case, it equals to the positive pairs (numerator part) that try to maximize, so it this seems to conflict with the intent.
6.	Line 485, different reconstruction strategies (categories and reconstruction target of Table 5) are confusing. It is unclear how different options for reconstructing trajectories starting at t' are justified or implemented. What do you mean by historical information for the sub-scenarios of t'? Could you clarify these strategies?

---

> ### Author Response · Authors · 2024-11-21
> **Response to Reviewer SCK1 (1/3)**
>
> Dear reviewer `SCK1`, we sincerely appreciate the detailed assessment and valuable feedback on our paper! Here, we provide responses and explanations to your comments and suggestions.
>
> > in the pretraining phase, it is unclear how the values for t and t' are selected for each dataset to avoid overlapping for the experiments.
>
> Thanks for the careful review! We avoid overlapping of t and t’ by enforcing different sampling ranges when we sample them. For example, in Argo dataset, the trajectory horizon is 50, with 20 as input and 30 as output, where we need to sample sub-scenarios with 20 timesteps. During sampling, we sample t within the range [0, 10] and t' within the range [t+20, 30] so that the two sub-scenarios have no overlapping timesteps. The same strategy, with varied sub-scenario lengths, is applied to other datasets such as Argo 2 and WOMD, to ensure no overlaps.
>
> > The horizons of the sub-scenario as input and reconstruction are also not specified; it would be helpful to know if these are consistent with the motion prediction settings (either input or output horizons?)  used during fine-tuning.
>
> Thanks for the question. As we briefly mentioned in lines 251 and 252 in our original paper (now they appear in lines 255 and 256 in our revised paper), to enable better alignment between pre-training tasks and the actual downstream prediction task, both the horizon of the input and reconstruction sub-scenario is designed to be consistent with the input horizon of the target downstream dataset.
>
> > Additionally, the default lambda value for the loss function is not provided.
>
> Thanks for pointing out! The default lambda value is set to 1. We've added this information to line 330 in our revised paper.
>
> > Will the code be made available as open-source?
>
> Certainly! It is a great pleasure to share our work with the community and contribute to advancing the foundation model in motion prediction. As part of our commitment to openness and collaboration, we plan to open-source our code upon the acceptance of the paper. To facilitate early access and feedback, we created an anonymous code repository during the rebuttal stage. This repository, though a rough and initial version for a preview, includes both pre-training and fine-tuning code, as well as our model checkpoints for pre-training and fine-tuning. You can access it through this [URL](https://anonymous.4open.science/r/5f404fda8de3e3278e2f794f80bffed0036c827b).
>
> > Computational cost is not shown and compared.
>
> Thanks for this constructive suggestion. The computation cost during pre-training highly depends on the datasets used. Argo 1, Argo 2, and WOMD provide 205k, 200k, and 487k training scenarios respectively, and all datasets add up to 900k scenarios. Pre-training our model on a single dataset typically takes 1~2 days for 128 epochs with 8 GPUs, and pre-training on multiple datasets can have prolonged training time. Note that for model parameters, our pretraining only introduced a few new MLPs to the original model.
>
> Regarding comparison to other SSL methods, as mentioned in our paper, to the best of our knowledge, only Forecast-MAE was open-sourced at the time of paper submission. In Forecast-MAE, the pre-training takes 1~2 days for 60 epochs with 4GPUs. As shown in the table below, we compare our SSL strategy with Forecast-MAE. When using the same number of pre-training epochs, our method outperforms Forecast-MAE, and extending the training epochs further enhances our performance, demonstrating the effectiveness of our method.
>
> | Pre-Training Method | Pre-Training Epochs | Backbone Model | minFDE | minADE | MR    |
> | ------------------- | ------------------- | -------------- | ------ | ------ | ----- |
> | \                   | \                   | Forecast-MAE   | 1.436  | 0.811  | 0.189 |
> | Forecast-MAE        | 60                  | Forecast-MAE   | 1.409  | 0.801  | 0.178 |
> | SmartPretrain       | 60                  | Forecast-MAE   | 1.394       |  0.796      |  0.174  |
> | SmartPretrain       | 128                 | Forecast-MAE   | 1.372  | 0.786  | 0.169 |

---

> ### Author Response · Authors · 2024-11-21
> **Response to Reviewer SCK1 (2/3)**
>
> > Clarifications on which parts of each network are used for pretraining would be beneficial (refer to question 1 below).
> >
> > In Fig 2, the figure labels the component as "model" for pretraining. However, since Section 3.1 (problem formulation) indicates that the contrastive loss is calculated on the encoded embeddings, should this component be referred to as the "encoder" instead? In the experiments with HiVT, HPNet, QCNet, and Forecast-MAE, are only the encoders of these models pretrained? For models like HPNet, which do not clearly differentiate between encoder and decoder architectures, how do you determine which parts of the model to use for pretraining to obtain latent embeddings? How might this selection influence the results?
>
> Thanks for the insightful and very detailed observation! As you point out, the choice of which part of the network is used for pre-training is related to the architecture of the model. We’ll introduce them with three categories:
>
> 1. For models with clear encoder and decoder architecture (HiVT, Forecast-MAE), we use the agent embeddings before the decoder, and only pre-train the encoder.
> 2. For models with refinement modules (QCNet, HPNet), the model consists of 1) an initial prediction stage with standard encoder-decoder architecture and 2) a refinement prediction stage. We only pre-train the encoder in the initial stage, since the training of the refinement module necessitates predicted trajectory while our pre-training tasks do not provide these predictions.
> 3. Models with special designs: HPNet incorporates a specially designed historical prediction mechanism, enabling predictions not only from the current time step but also from historical time steps. During pre-training, we explored two approaches: 1) the standard approach: using agent embeddings only from the current time step for the SSL task; 2) the HPNet-adapted approach: using agent embeddings from all historical time steps for the SSL task, and averaging the loss from all historic time steps. Interestingly, we observed similar performance between these two approaches. This outcome is likely attributed to our temporal sampling strategy, which effectively captures and integrates temporal information.
>
> In summary, “model encoder” is indeed a more accurate name than “model”, we have modified this in Fig.2 of our revised paper.
>
> > line 95-97, I find this claim unclear and potentially misleading. I disagree that MAE pretraining is inflexible; on the contrary, I think the masking pretraining is quite versatile. The masking techniques used in the papers you referenced - Rmp, Traj-mae, Forecast-mae, Sept - appear very similar, suggesting the masking concept is not limited and can be readily applied across various models.
>
> Thanks for pointing out this. Our statement in Line 95-97 may have not been accurate. We do agree that MAE pre-training is flexible, when they are applied on agent trajectory reconstruction. The point we want to emphasize is that the MAE approach based on map reconstruction is not general, since 1) many works focus on aggregating agent embeddings and provide explicit access to them, while explicit map embeddings are not always available (for example, HPNet, QCNet and some other GNN-based models); 2) different works may take different map representations, such as a vectorized map and rasterized map, thus the map reconstruction pretraining strategy need to be designed for each representation and could be less general. We have updated the claim and made it more clear in line 97-100 of our revised paper.
>
> > Line 104, abbreviation CL is not explained before.
>
> Thanks for pointing out. By CL we refer to contrastive learning. We have added the full name before CL to clarify it in line 102 of our revised paper.

---

> ### Author Response · Authors · 2024-11-21
> **Response to Reviewer SCK1 (3/3)**
>
> > Line 266 mentions that only complete trajectories are used in pretraining, excluding incomplete ones. During pretraining phase, do you reconstruct single-agent trajectory or multi-agent? The same question applies to the prediction phase. Could you specify what percentage of trajectories remain after filtering for each dataset?
>
> Thanks for the insightful question. Regarding single-agent or multi-agent pre-training, we follow the backbone model's original training setting. Specifically, multi-agent training has become popular in recent literature, since it can be seen as a data augmentation strategy to enhance data diversity and model performance. All four backbone models considered in our experiment adopt multi-agent training, thus our pre-training did the same. Regarding the downstream prediction training phase, again we follow the backbone models’ original setting and adopt multi-agent training.
>
> The percentage of after-filtering complete trajectories, over all vehicle trajectories is 35%, 27% and 25% for Argo, Argo2 and WOMD respectively.
>
> > Line 296, for eq.(1), why the case of i=j is not excluded in the second term of the denominator, as in this case, it equals to the positive pairs (numerator part) that try to maximize, so it this seems to conflict with the intent.
>
> Thanks for the detailed question! We follow classical contrastive learning methods (i.e., SimCLR, MoCo), to consider all data pairs, including the positive pair, in the denominator, so that the denominator serves as a normalization term. This ensures that the loss function is mathematically consistent and effectively balances the contributions of positive and negative pairs. We’re also happy to provide more discussion/information regarding the design of Eq.1, if needed.
>
> > Line 485, different reconstruction strategies (categories and reconstruction target of Table 5) are confusing. It is unclear how different options for reconstructing trajectories starting at t' are justified or implemented. What do you mean by historical information for the sub-scenarios of t'? Could you clarify these strategies?
>
> Thanks for pointing this out and we are happy to clarify these ablation strategies. Table 5 aims to ablate the influence of different reconstruction targets, on the downstream prediction performance. In Table 5, the first row represents the variant where we do not conduct reconstructive pretraining, and only contrastive pretraining is considered. The second and third rows belong to one category "reconstruction with historical information", which means the trajectories of the input sub-scenario t are included in the reconstructed trajectories. For example, the reconstruction target of row 2 is set as exactly the trajectories of the sub-scenario t, which forms a self-reconstruction task. Row 3 reconstructs the trajectories of the entire scenario (e.g., 20 points as input to reconstruct all 50 points in Argo). The last two rows represent the category "reconstruction with predictive information", which means we don't include trajectories of the sub-scenario t in our reconstruction targets, but instead, aim to predict the remaining trajectories. In row 4, we reconstruct the complementary trajectory of the input sub-scenario (20 points as input and complementary 30 points for reconstruction in Argo). Row 5 aims to reconstruct the trajectory of the other sub-scenario (sub-scenario t').
>
> The results of Table 5 indicate that the last two reconstruction strategies show the biggest performance boost, and we choose row 5 as our final actual model variant.

---

> > ### Comment · Reviewer_SCK1 · 2024-12-02
> > **Thanks for the rebuttal**
> >
> > Thank you very much for your efforts in providing clarifications and conducting additional experiments. Your responses address most of my concerns and questions. I have reviewed all the reviewers' comments and your replies. I have just a few comments below. I will maintain my current score.
> >
> > $\ $
> >
> > > (Clarification on Table 5) The last two rows represent the category "reconstruction with predictive information", which means we don't include trajectories of the sub-scenario t in our reconstruction targets, but instead, aim to predict the remaining trajectories. In row 4, we reconstruct the complementary trajectory of the input sub-scenario (20 points as input and complementary 30 points for reconstruction in Argo). Row 5 aims to reconstruct the trajectory of the other sub-scenario (sub-scenario t').
> >
> > Thanks for the detailed explanations for Table 5, I do think a clearer elaboration in the final version is needed. I understand the table much better now, for Argo dataset, for instance, input is first [t, t+20], the reconstructed output from rows 2 to 5, represent to [t, t+20], [t, t+50], [t+20, t+50], [t', t'+20], respectively.
> >
> > Interestingly, row 4 (complementary trajectory of the input sub-scenario, basically motion prediction of t) and row 5 (trajectory of the other sub-scenario t' input, one of the core things of this paper) show relatively equivalent good performance in the ablation results. What could this imply? Does it suggest that pretraining with motion prediction tasks alone is sufficient for achieving strong performance on a single dataset?
> >
> >
> > > (regarding the statement in lines 95-97) The point we want to emphasize is that the MAE approach based on map reconstruction is not general.
> >
> > > "MAE approaches demand that each trajectory and map segment must have an explicit feature representation to enable reconstructive pre-training" (line 96, the revised version)
> >
> > The sentence you provided in the rebuttal makes some sense. However, the description in line 96 is not accurate. MAE is a general framework, and it can be applied to tasks like masking and reconstructing trajectories only, as shown in methods like Rmp. I recommend revising this line.
> >
> >  > The percentage of after-filtering complete trajectories, over all vehicle trajectories is 35%, 27% and 25% for Argo, Argo2 and WOMD respectively.
> >
> > I hope this important statistic and the details of how non-overlapping t and t' trajectories are sampled for each dataset, can be included in the appendix of the final version.

---

> > > ### Author Response · Authors · 2024-12-03
> > >
> > > Dear Reviewer `SCK1`:
> > >
> > > Thank you for your feedback and for maintaining your current score. We address each of your new comments as follows:
> > >
> > > > Interestingly, row 4 (complementary trajectory of the input sub-scenario, basically motion prediction of t) and row 5 (trajectory of the other sub-scenario t' input, one of the core things of this paper) show relatively equivalent good performance in the ablation results. What could this imply? Does it suggest that pretraining with motion prediction tasks alone is sufficient for achieving strong performance on a single dataset?
> > >
> > > Thanks for the question. All results shown in Table 5 are pre-trained with our contrastive learning task as well, and we have adapted different reconstruction targets based on it. We present the performance of only doing the reconstruction learning task in Table 4, which indicates: 1) the reconstruction task alone can effectively improve prediction accuracy in isolation, and 2) combining both tasks yields the largest improvement.
> > >
> > > Also, as inspired by Reviewer `TuBk`, we've added an experiment using the standard motion prediction task as the pre-training task. The results are relatively poor compared with the SSL task (seen in our responses to Reviewer `TuBk`).
> > >
> > > > The sentence you provided in the rebuttal makes some sense. However, the description in line 96 is not accurate. MAE is a general framework, and it can be applied to tasks like masking and reconstructing trajectories only, as shown in methods like Rmp. I recommend revising this line.
> > >
> > > Thanks for the careful review. We will revise it to make our claim more accurate in the final version. Specifically, we will separate the MAE approach with/without map reconstruction and address the “masking and reconstructing trajectories only” method like Rmp.
> > >
> > > > I hope this important statistic and the details of how non-overlapping t and t' trajectories are sampled for each dataset, can be included in the appendix of the final version.
> > >
> > > Thank you for this suggestion. We agree that these details are important to fully understanding the method design and will add them to the appendix in the final version.

---

### Author Response · Authors · 2024-11-27
**Revised Manuscript Submission**

Dear AC and Reviewers,

We would like to sincerely appreciate the time and effort you have invested in reviewing our submission. As the “uploading a revised PDF” phase is drawing to a close (November 27th), we’ve uploaded a revised version of our manuscript. Two minor changes are made: 1) we updated a few claims (marked as blue in the main text) based on the feedback from all Reviewers in our responses; 2) we removed Fig. 4 (visualization results) from the main text and incorporated it with additional visualization results in Appendix A.1 (following the suggestions of  Reviewer `wPEn` and `TuBk`) to ensure the main text adheres to the 10-page limit.

We are continuing to polish our manuscript after this deadline to incorporate all new results and discussions with reviewers. Thank you once again for your thoughtful review and consideration.
Looking forward to your further responses and comments. We are more than happy to provide any further details or explanations.

Best regards,

The Authors

---

### Meta-Review · Area_Chair_ZSWd · 2024-12-23

**Metareview:**

This paper proposes a pretraining framework for trajectory prediction tasks using real-world datasets. Key ideas include self-supervised pre-training methods (contrastive and reconstruction learning). Experiments is performed on multiple datasets including Argoverse 1/2, Waymo Open Motion Dataset and a series of ablation studies demonstrate the approach.

Most of the reviewers are positive about the paper. The most critical review points out that the methods have been used in other domains such as NLP, however application to this domain is novel. Consequently, in light of extensive experimentation, while being incrementally innovating my recommendation is accept as a poster.

**Additional Comments On Reviewer Discussion:**

Majority of the reviews are positive and the most critical review did not show strong conviction towards arguing against accepting the paper.

---

### Decision · Program_Chairs · 2025-01-22

Accept (Poster)